# A dataset of standard precipitation index reconstructed from multi-proxies over Asia for the past 300 years

Yang Liu[1], Jingyun Zheng[1,2], Zhixin Hao[1,2], Quansheng Ge[1,2]

[1] Key Laboratory of Land Surface Pattern and Simulation, Institute of Geographic Sciences and Natural Resources Research, Chinese Academy of Sciences, Beijing 100101, China
[2] College of Resources and Environment, University of Chinese Academy of Sciences, Beijing 100049, China

*Correspondence to*: Jingyun Zheng (zhengjy@igsnrr.ac.cn); Quansheng Ge (geqs@igsnrr.ac.cn)

**Abstract.** Proxy-based precipitation reconstruction is essential to study the inter-annual to decadal variability and underlying

mechanisms beyond the instrumental period that is critically needed for climate modeling, prediction, and attribution. Based on 2912 annually resolved proxy series mainly derived from tree-ring and historical documents, we present a set of standard precipitation index (SPI) reconstructions for annual (Nov-Oct) covering entire Asia and for wet season (i.e., Nov-Apr for western Asia and May-Oct for the others) with the spatial resolution of 2.5° since 1700. To screen the optimal candidate proxies for SPI reconstruction in each grid from available proxies in its connected region with a homogeneous rainfall

regime and similar precipitation variability, a new approach is develop by adopting the grid-location-dependent division derived from the instrumental SPI data. The validation shows that these reconstructions are effective for most of Asia. The assessment of data quality compared with gauge precipitation before calibration time indicates that our reconstruction has high quality to show the precipitation variability in most of the study areas except for few grids in western Russia, the coastal area of southeast Asia and northern Japan.

## 1 Introduction

Asia bears the brunt of flood and drought disasters associated with extensive social and economic damages than any other continent due to its large and heterogeneous landmass, plus high population densities in the south and east regions (Lee et al., 2020; Wei et al., 2020). At the national level, seven of the top ten countries in the world with the largest number of

population affected by climate-related disasters (mainly flood and drought) are located in Asia (CRED and UNISDR, 2015). However, the inter-annual, decadal and centennial spatiotemporal variability of Asian precipitation and underlying mechanisms have not been fully characterized, which limits the performance of precipitation projection for the next decades to hundred years (Seth et al., 2019; Wang et al., 2021; Liu et al., 2022a). Long-term, spatially-resolved, and high-quality precipitation datasets are needed to address these issues. Unfortunately, the global precipitation observation network only

covers the past century (Sun et al., 2018) while the data for the first half period in Asia is at low confidence levels (Hartmann et al., 2013). Therefore, proxy-based precipitation reconstructions are essential to quantify the precipitation variability beyond the instrumental period.

Up to now, there are four gridded datasets to reconstruct summer (or the warm season) precipitation variability in mid-low latitude Asia for the past hundreds of years (Cook et al., 2010a; Feng et al., 2013; Shi et al., 2018; Shi et al., 2017), by using tree-ring chronologies only or merging multi-proxies. For example, using 327 tree-ring chronologies mainly located in the Tibetan Plateau and Mongolia, Cook et al. (2010a) reconstructed the gridded (2.5° × 2.5°) summer (Jun–Aug, JJA) Palmer drought severity index (PDSI) over Monsoon Asia during 1300-2005. By weighted merging 453 tree-ring-width chronologies and 71-site dryness/wetness grade series derived from Chinese historical documents (local gazettes), Shi et al. (2018) reconstructed a gridded Asian summer precipitation dataset for 1470-2013. Similar reconstructions were also conducted for North America (Cook et al., 2010b; Stahle et al., 2020), Europe (Cook et al., 2015; Cook et al., 2020), and Oceania (Palmer et al., 2015). Moreover, by using a data assimilation (DA) approach to combine 2978 proxy data with the physical constraints of the atmosphere-ocean climate model together, a globally gridded (2.0° × 2.0°) hydroclimate index dataset over the Common Era was also reconstructed (Steiger et al., 2018), including PDSI and the standardized precipitation evapotranspiration index (SPEI) for JJA, DJF (Dec-Feb) and April to the next March. These datasets extend records back in time and provide valuable efforts on improving the gridded paleoclimate reconstruction by synthesizing multi-proxies from individual sites with spatiotemporal inhomogeneity.

However, intercomparisons of the abovementioned four gridded precipitation/drought variability reconstructions in monsoon Asia (Cook et al., 2010a; Feng et al., 2013; Shi et al., 2018; Shi et al., 2017) with independent instrumental observation data show notable differences among them caused by proxies and methods for calibration, particularly dominated by the number and sample distribution of proxies used, as well as the seasonal sensitivity of the individual proxy to precipitation anomaly (Liu et al., 2021). For example, in the reconstruction only from tree-ring proxies, the explained variance in regions with spars proxies (e.g., eastern China, Mainland Southeast Asia) is usually less than 20% (Cook et al., 2010a). By merging tree-ring and documentary proxies in the reconstruction, the result is believed to illustrate large-scale rainfall variability faithfully but has more uncertainties in representing regional rainfall anomalies (Shi et al., 2018). Moreover, the precipitation over Asia has a complex spatial pattern with the temporal variability on intra-seasonal and inter-annual scales (Hsu et al., 2014) due to different rainfall regimes in space (Awan et al., 2015; Conroy and Overpeck, 2011). Therefore, the sensitivity of individual proxy to precipitation anomaly has evident regional differences for seasons. In addition, many new proxies achieved in recent years are not utilized in the above-mentioned four gridded reconstructions in monsoon Asia (e.g., Shah et al., 2007; Sass-Klaassen et al., 2008; Arsalani et al., 2018; Arsalani et al., 2015; Chen et al., 2016; Zhang et al., 2017; Pumijumnong et al., 2020; Xu et al., 2015; Buckley et al., 2017; Ukhvatkina et al., 2021; Akkemik et al., 2020; Kostyakova et al., 2017; Kucherov, 2010; Xu et al., 2013; Borgaonkar et al., 2010). All of these motivate us to initiate this new gridded (2.5° × 2.5°) reconstruction effort on seasonal to annual precipitation variability over the past 300 years in the whole Asia including the western and northern Asia not covered in four gridded datasets developed in previous

studies (Cook et al., 2010a; Feng et al., 2013; Shi et al., 2018; Shi et al., 2017). Noted that most of Russian territory is located in Asia, to keep the data integrity at the national level, the whole Russian territory is included in this study.

## 2 Data and Method

### 2.1 The study area and the framework for grid SPI reconstruction

The spatial coverage of our reconstruction is shown in Fig. 1, and the reconstructed target is standard precipitation index (SPI). In this vast study area, there are many climatic types with heterogeneous precipitation, specifically, the wet season in western Asia and the southwest part of central Asia is mainly from November to April (Nov-Apr), but that in the rest regions is May-October (May-Oct) due to different rainfall regimes in different regions (Bombardi et al., 2019; Peng et al., 2020). Thus, we reconstructed the annual (November to October, Nov-Oct) SPI for the entire study area, as well as the Nov-Apr SPI in western Asia and the southwest part of central Asia, and the May-Oct SPI in the other regions for the wet season, respectively. The flow-chart of the reconstruction procedures is shown in Fig. 2. Noted that there exists a complex spatial coherence pattern for the precipitation variation on scales of inter-annual, decadal and longer in the study area, which means the spatial representativeness of the individual proxy is dominated by the location in the context of the region (e.g., shape and area) with coherent rainfall regime and variation. Therefore, we develop a new approach to select proxies for each grid SPI reconstruction by adopting the grid-location-dependent division (GLDD) derived from the instrumental SPI data, instead of selecting proxies usually from an isotropic search radius for all grids in many previous studies (e.g., Cook et al., 2010a; Shi et al., 2018). The methods for reconstruction, including GLDD, searching candidate proxies for calibration and validation will be presented in Section 2.4.

### 2.2 Instrumental data for calibration and spatial pattern of wet season identification

In our study, the grid size for SPI reconstruction is set as $2.5^{\circ} \times 2.5^{\circ}$. The instrumental data used for calibration is resized from the $0.5^{\circ} \times 0.5^{\circ}$ gridded monthly SPI data for 1948-2019 calculated by NOAA's land precipitation product (Chen et al., 2002), which was downloaded via IRI/LDEO Climate Data Library (http://iridl.ldeo.columbia.edu/SOURCES/.IRI/.Analyses/.SPI/). As pointed out by previous studies (Bombardi et al., 2019; Peng et al., 2020; Nieto et al., 2019), moisture sources are different across Asia throughout the year and the wet season could be roughly classified as two terms of Nov-Apr and May-Oct. Therefore, to identify the spatial pattern of the wet season for SPI reconstruction in the $2.5^{\circ} \times 2.5^{\circ}$ gridded scale induced by different regional rainfall regimes, the monthly precipitation data for 1948-2019 by GPCC (Schneider et al., 2017) is also used to calculate two consecutive months with the most rainfall amount in a year (Fig. 1). It is shown that in most parts of the study area, the wettest two consecutive months are in May-Oct. However, in western Asia (excluding the south corner of the Arabian Peninsula), the southwest part of central Asia, and the

tropical zone south to 10°N, the wettest two consecutive months are in Nov-Apr. Moreover, there also exists a few grids (dot
marked in Fig. 1) that have no distinct wet season (Bombardi et al., 2019). Thus, we exclude the dotted grids in wet season
SPI reconstruction.

## 2.3 Proxy data preparing

There are a total of 2912 annually resolved proxy series from Asia and adjacent land areas (Eastern Europe and Alaska)
for reconstruction, of which 2792 are derived from tree-ring, 115 from historical documents, 4 from ice cores, and 1 from a
stalagmite. Their spatial and temporal distribution is shown in Fig. 3. Noted that all of the proxy series have at least 20
records overlapped with the instrumental period since 1948 to ensure a sufficient sample size for calibration and validation,
and more than 30 records before 1948 for reconstruction. The data source and standardized processes for each type of proxy
series are described below.

Tree-ring data are mainly (2772) from the International Tree Ring Data Bank (ITRDB), maintained by the World Data
Center for Paleoclimatology (WDC-P, https://www.ncei.noaa.gov/products/paleoclimatology), including 1854 tree-ring
width records, 828 tree-ring density records, 67 tree-ring latewood percent records, 22 tree-ring stable oxygen isotope ($\delta^{18}$O)
and 1 tree-ring blue intensity record. Most sites have two categories of data, i.e., original raw tree-ring measurements and
tree-ring index chronologies derived from raw measurements. However, the index chronologies are not used directly in this
study because they were standardized by various methods which are not described in the online metadata and some of the
methods may result in a substantial loss of long-term fluctuations (Coulthard et al., 2020). To maximally preserve the
climatic related low-frequency variance, we recalculate the chronologies from 2644 available raw measurement files by
removing the growth trend with age-dependent splines (Melvin et al., 2007). In a few cases where age-dependent splines
contain zeros or negative values, a more flexible curve, Friedman variable span smoother (Friedman, 1984), is used to fit the
growth trend. In addition, some trees experience disturbances during their lifespan, which could cause abrupt growth
increases or reductions (Altman, 2020). To eliminate this effect, the running mean technique (Altman et al., 2014) is applied
to identify the disturbance event then separate growth curves are fitted before and after this year. Finally, the 51-year sliding
expressed population signal (EPS) is calculated and the threshold of 0.85 is used to determine the first reliable year of a
chronology. The above procedures are also applied for sites with raw measurements only. The other 128 tree-ring records
from ITRDB only have chronologies, EPS is not available thus we use the minimum sample size of 5 to determine the first
reliable year. Besides ITRDB, 17 tree-ring width chronologies and 3 tree-ring $\delta^{18}$O chronologies that indicate local
precipitation or drought from recently published papers are included in our study (Shah et al., 2007; Sass-Klaassen et al.,
2008; Arsalani et al., 2018; Arsalani et al., 2015; Chen et al., 2016; Zhang et al., 2017; Pumijumnong et al., 2020; Xu et al.,
2015; Buckley et al., 2017; Ukhvatkina et al., 2021; Akkemik et al., 2020; Kostyakova et al., 2017; Kucherov, 2010; Xu et
al., 2013; Borgaonkar et al., 2010). Compared with the tree-ring network used in previous studies over the monsoon Asia
region (Cook et al., 2010a; Feng et al., 2013; Shi et al., 2018; Shi et al., 2017), a total of 113 ring-width chronologies are
added in our study.

It is worth noting that part of sites consist both tree-ring width and density chronologies. According to the principle of dendroclimatology, the availability of soil water affects the growth rate and formation of wood, both within a season and the longer terms, thus, tree-ring width is expected to be positively correlated with precipitation via this direct response (Vaganov et al., 2011; Wettstein et al., 2011). For tree-ring density chronologies, they are usually correlated with temperature variation and scarcely used in precipitation reconstruction (Briffa et al., 2002). However, due to multiple types of climate and complex topography in the vast study area, the tree-ring density chronologies and width chronologies with negative correlations to precipitation may also well indicate precipitation variation (George, 2014) and the use of such tree-ring predictors in hydro-climate reconstruction has been discussed in prior studies (e.g. Cook et al., 2020). Therefore, we reconstruct two versions of SPI, one excludes tree-ring width chronologies negatively correlated to precipitation and tree-ring density chronologies (hereafter called as "Version A"), the other include all tree-ring chronologies (here after called "Version B").

The proxy from historical documents is mainly the dryness/wetness grade series for 120 sites in China for the past 500 years (henceforth referred to as DW120) by Chinese Academy of Meteorological Science (CAMS, 1981). The grades were calibrated based on descriptions of drought/flood and their impacts during the wet season mainly recorded in Chinese local gazettes using ideal frequency criteria of all time, roughly 10% for grades 1 and 5 (heavy flood and severe drought), 20-30% for grades 2 and 4 (flood and drought), and 30-40% for grade 3 (normal). This grade dataset originally ended in 1979 (CAMS, 1981) and was extended to 2000 (Zhang et al., 2003; Zhang and Liu, 1993), which becomes an essential dataset to reconstruct summer precipitation over the Asia monsoon domain (Feng et al., 2013; Shi et al., 2018; Shi et al., 2017). However, DW120 contains a large proportion of missing data because there are only 26040 grade records since 1700 (Fig. 4a), which limits the spatial and temporal coverage of data for grid SPI reconstruction. Therefore, we update this dataset by two steps. The first is adding the missing data in DW120 from another dryness/wetness grade dataset for 63 sites in central eastern China (DW63) developed by Zhang (1996). Noted that all of sites of DW63 are included in DW120 and the grading criteria for DW63 is same as DW120. Since DW63 was reconstructed from more abundant historical documents (such as the drought/flood descriptions recorded in the memoirs and archives of the Qing Dynasty), it had fewer missing records with 100% data availability after 1700. Therefore, all missing records of DW120 in central eastern China are added from DW63, which supplements 2045 grade records in total. The second is interpolation from the isoline map of DW120 for individual years when most sites have available data (CAMS, 1981), which supplements 4121 grade records. Since DW63 was reconstructed by the same grading criteria as DW120, both of the added 2045 grade records from DW63 and the added 4121 records from the yearly isoline map of DW120 match with the original available data. This updated DW120 finally contains 32206 grade records since 1700, which has a 23.7% increase compared with the original version (Fig. 4b). Unfortunately, no data are available before the 20th century for 10 sites in west China and one site in northeast China (cross marked in Fig. 4b), thus only 109 sites in China are selected for our SPI reconstruction. Another documentary-based dryness/wetness grade series since 1781 is from Mumbai, India, which also consists of 5 grades calibrated against the percentage of rainfall anomaly derived from instrumental data in their overlapped period (Adamson and Nash, 2014). In addition, the series of wet-season (May to October) rainy day for 5-site in Japan are also included. These series were extracted from the historical

diaries (https://www.ncei.noaa.gov/access/paleo-search/study/5412) and merged with instrumental data (Kamiguchi et al., 2010) by the method from Murata (1992).

The rest proxy series derived from 4 ice cores in the Himalayas and one stalagmite in India, are also downloaded from WDC-P and have been proved to indicate hydro-climatic change by prior studies (Thompson et al., 2000; Sinha et al., 2011; Qin et al., 2002). It is worth noting that the $\delta^{18}O$ ratio series of the ice core from East Rongbuk Glacier is unequally spaced with a mean temporal resolution of 0.082 years, which is simply re-sampled to an annual resolved series by averaging data in the same year.

## 2.4 Method for grid SPI reconstruction with grid-location-dependent division

Since there are many climatic types with heterogeneous precipitation in the study area and the spatial representativeness of individual proxy is sensitive to location, we develop a new approach to identify the region for searching proxies (called "searching region" hereafter) to reconstruct SPI in each grid. This approach is developed according to the regional division of the coherence of inter-annual precipitation variations in context of the spatial pattern of rainfall regime, to ensure the proxies in the searching region can well indicate SPI variability in the target grid. We divided the regions from the spatial pattern of correlation coefficient (CC) between SPI of each target grid for SPI reconstruction and the other grids within study area, which is calculated from instrumental SPI data. The searching region is defined as all connected grids surrounding the target grid with CC passing 0.05 significant level. Thus, this searching region has the robust coherence of precipitation variability and rainfall regime with the target grid, and the proxies in this region have best spatial representativeness to target grid. Since this regional division is dependent on the grid-location by rolling the target grid, this approach was called "grid-location-dependent division (GLDD)" in our study (Fig. 2). Moreover, we use best subset regression (BSR) to identify optimal combinations of the candidate proxies for calibration in each grid SPI reconstruction based on available proxies in different intervals respectively (Fig. 2), because the proxies are uneven distributed in space and time. Following shows the Nov-Oct SPI reconstruction for a grid of 90.0-92.5°E and 27.5-30.0°N (located in southwest China) as an example for the detailed steps.

Firstly, calculating the spatial SPI correlation field of the target grid and identifying the regions with positively significant ($p<0.05$) correlation coefficients (Fig. 5a). It shows that the target grid and its adjacent grids have significant correlations that cover an irregular shape (i.e., not a circle-like with isotropic radius from the target grid or other regular shape) in southwest China, which means there exists robust coherence for SPI variation. This is because the rainfall regime and precipitation for that region are usually dominated by the same atmospheric circulation systems (Zhang and Wang, 2021). Besides, there are some other remote regions (e.g., the Malay Archipelago, Russian Plain, and regions around New Siberian Islands) that show significant correlations to the target grid. However, prior studies had reported that the long-distance precipitation teleconnection patterns are usually unstable through a long-term period since they are linked by large-scale atmospheric circulations or propagating waves (Wu, 2016; Boers et al., 2019). Therefore, the candidate proxies for the

target grid SPI reconstruction should be searched only from the connected region (Fig. 5b), and there are a total of 43 proxies for the candidate proxy selection.

Secondly, calculating the correlations between the target grid SPI and each series of 43 proxies in the searching region to select the candidate proxy by the threshold of the 0.1 significance level for the correlations. Note that prior summer precipitation could affect the tree-ring formation in the next year (Wettstein et al., 2011), thus 1-year lagged tree-ring chronologies are also included for the Nov-Oct SPI reconstruction. However, the proxies with highly positive correlations may lead to multi-linearity effects in the regression equation for calibration. Thus, we also calculate the correlations among

all 43 proxy series, and if any pair of proxy series shows an extremely high positive correlation (i.e., $r > 0.90$ and $p < 0.0001$) in their common period, the shorter one will be excluded from the pool of candidate proxies. By this step, a total of 8 proxy series (including 5 tree-ring width series, 1 tree-ring $\delta^{18}O$ series, and 2 dryness/wetness grade series) are selected for BSR in the following step (Fig. 5b).

Thirdly, establishing the calibration equation by using BSR for each time segment depends on the length of the

candidate proxy series. According to the start and end year of all 8 candidate proxy series, the time of proxy availability should be classified into 6 segments, in which there are 8 candidate proxies for 1772-1997, 7 candidate proxies for 2 segments in 1745-1771 and 1998-2000 respectively, 6 candidate proxies for 1743-1744, 5 candidate proxies for 1739-1742, and 4 candidate proxies for 1700-1938 (Fig. 5c). Moreover, to avoid the overfitting in the regression induced by redundant independent variables (Lever et al., 2016), if there are more than 5 candidate proxies, only 5 proxy series with top 5

significant levels for the correlations with target SPI are retained for the regression. This is because that the sample length to develop calibration equations for reconstruction is about 50 years usually, and the sample size should be preferably 10 times (or more) the number of variables for BSR according to the principle of statistics (Sekaran, 2003). Thus, 3 individual segments (1743-1744, 1745-1771 and 1772-1997) retain the same 5 proxies (i.e., two tree-ring width series, one tree-ring $\delta^{18}O$ series and two DW120 series), and they could be regarded as one segment of 1743-1997. Then, we use BSR to establish

4 calibration equations for SPI reconstruction in 1700-1738, 1739-1942, 1943-1997, 1998-2000 respectively, in which the best subset selection is determined by maximizing the Coefficient of Efficiency (CE) (Cook et al., 1994) calculated by a state-of-the-art 4-fold rolling window cross-validation procedure (Nguyen et al., 2020). Another commonly used validation parameter, reduction of error (RE), is also calculated from the same procedure. Finally, the target SPI series for the full time is constructed by merging the reconstructions for individual segments (Fig. 5d). As the reconstructions for different segments

were calibrated from different equations with different variances and predicted sums of squares, the magnitudes of the reconstructed SPI for a specific segment had to be adjusted using the variance matching method with respect to the standard deviations of the predictands in common years during the calibration period (McCarroll et al., 2015).

# 3 Results and discussion

The dataset of includes 4 SPI reconstructions: (1) the Nov-Oct SPI reconstruction for entire Asia without using tree-ring density chronologies and width chronologies with negative correlations to precipitation (Nov-Oct SPI Version A); (2) the Nov-Oct SPI reconstruction for entire Asia by adding tree-ring density chronologies and width chronologies with negative correlations to precipitation (Nov-Oct SPI Version B); (3) the wet season SPI reconstruction for the extra-tropical Asia (Nov-Apr SPI for western Asia and May-Oct SPI for the rest regions) without using tree-ring density chronologies and width chronologies with negative correlations to precipitation (wet season SPI Version A); (4) the wet season SPI reconstruction for the extra-tropical Asia (Nov-Apr SPI for western Asia and May-Oct SPI for the rest regions) by adding tree-ring density chronologies and width chronologies with negative correlations to precipitation (wet season SPI Version B). Each of them is stored in a NetCDF file (.nc) and contains 5 three-dimension (longitude × latitude × time) variables, including reconstructed SPI, adjusted coefficient of determination ($R^2a$), validation RE, validation CE and the number of proxies used for construction (nPrx).

## 3.1 Validity of the reconstruction

Figures 6 show the spatial patterns of R2a, RE and CE for the Nov-Oct SPI reconstruction since 1700 by a 50-year interval. It shows that CE in the most of study areas is positive. Although few grids have negative CE, especially before 1800, most of them still have positive RE. These results mean the reconstruction is effective, in which the area with $R^2a > 0.2$ accounts for 36.1% of grids in Asia in 1700, and extends to 66.1% in 1950 due to more and more available proxies. Since 1700, the areas with $R^2a > 0.4$ are distributed in a board region from the southwest coast of the Caspian Sea to Balkhash Lake to eastern China, and some grids in the northern Far East, northern India and western Indochina Peninsula. $R^2a$ gradually passed 0.4 from 1750 to 1800 over Turkey, West Siberian Plain, central Asia, Mongolia and India. The highest $R^2a$ (more than 0.6) appeared in central eastern China throughout the entire 300-year period.

By adding tree-ring density chronologies and width chronologies with negative correlations to precipitation in the Nov-Oct SPI reconstruction (i.e. Version B), the number of grid with ineffective reconstruction is significantly reduced and the $R^2a$ for most of the grids is significantly increased (Fig. 7). Compare to the Nov-Oct SPI reconstruction Version A (Fig. 5), the area with $R^2a > 0.2$ in Version B accounts for 59.3% of grids in Asia in 1700, and extends to 85.1% in 1950. In particular, $R^2a$ increases by 0.2~0.3 in central to eastern Russia and by 0.1 to 0.2 in other regions except for the Arabian Peninsula and eastern China.

Likewise, for the wet season SPI reconstruction, it is also effective in most grids (Fig. 8), in which the area with $R^2a > 0.2$ accounts for 37.3% of grids in Asia in 1700, and extends to 61.5% in 1950. Compared with the Nov-Oct SPI Version A (Fig. 6), the wet season SPI reconstruction shows significantly higher $R^2a$ (0.1-0.2) for the region on the east of the Caspian Sea, slightly higher $R^2a$ (around 0.1) for most grids in high latitude zone, while a reduced $R^2a$ around 0.1 in eastern China. For the wet season SPI reconstruction by adding tree-ring density chronologies and width chronologies with negative

correlations to precipitation, the percentage of area with $R^2a > 0.2$ in 1700 and 1950 is 58.1% and 83.9%, respectively (Fig. 9). The difference in skill metrics between two wet season SPI versions (Fig. 8 and 9) is similar to that between two Nov-Oct SPI versions (Fig. 6 and 7).

## 3.2 Data quality and usability

Compare to three reconstructions of summer (JJA or May-September) precipitation (or PDSI) in monsoon Asia by
previous studies (Cook et al., 2010a; Feng et al., 2013; Shi et al., 2018), the $R^2a$ in the calibration period of our May-Oct SPI reconstructions (Fig. 8p and Fig. 9p) are 10% higher than that of the best one in three reconstructions over south Tibetan Plateau to eastern India subcontinent, western Indochina peninsula and northwest China. Moreover, our reconstruction has a slightly higher $R^2a$ in parts of Mongolia, central Asia and eastern China than that in other reconstructions. In particular, $R^2a$ in eastern China in our reconstruction is about 40% higher than that from the reconstruction only by tree-ring data (Cook et
al., 2010a). These improvements are not only because more proxy data (including the DWI derived from Chinese historical documents and the tree-ring data published recently) are added, but also because the development of the reconstruction method that selects proxies by the GLDD approach from a connected searching region with significantly positive correlations to the target grid SPI.

In addition, the maps of correlation between our wet season SPI reconstructions and four reconstructions in monsoon
Asia by previous studies show that most of grids pass 0.01 significant level (Fig. 10-11). Specifically, for the correlation (Fig. 10a, Fig. 11a) between our wet season SPI reconstruction Version A/B and the JJA precipitation reconstruction by Shi et al. (2018), there are 63.2%/64.1% of all grids passed the 0.01 significant level, in which the value of correlation coefficients for central eastern China are almost higher than 0.60. Similar results are also found for the correlations between our reconstruction versus the May-September precipitation anomaly reconstruction by Shi et al. (2017) in China (Fig. 10b, Fig.
11b) and the May-September precipitation reconstruction over monsoon Asia (Fig. 10c, Fig. 11c) by Feng et al. (2013). Even for the correlations between our wet season SPI reconstruction versus JJA PDSI reconstruction for monsoon Asian (Fig. 10d, Fig. 11d) by Cook et al. (2010a) only using tree-ring, 57.4% (for our reconstruction Version A versus JJA PDSI reconstruction) and 58.8% (for our reconstruction Version B versus JJA PDSI reconstruction) of all grids passed the 0.01 significant level.

To further assess the quality of reconstructed data, we compare our Nov-Oct SPI reconstruction with gauge precipitation at those weather stations with at least 30-year records before 1948, in which the precipitation data is from the Global Historical Climatology Network monthly dataset version 2 (GHCNmv2, https://www.ncei.noaa.gov/pub/data/ghcn/v2/) and Long-Term Instrumental Climatic Data Bases of the People's Republic of China (Tao et al., 1997). We calculate the correlations between Nov-Oct precipitation anomaly percentage and Nov-Oct SPI
reconstruction in corresponding grids (Fig. 12). Noted that the length of instrumental data before 1948 varies for different weather station, i.e., the degree of freedom for calculating these correlations are different station by station, we show the significant level for all positive correlations station by station instead of the value of correlation coefficient directly with

levels of $p \leq 0.01$, $0.01 < p \leq 0.05$, $0.05 < p \leq 0.1$ and $p > 0.1$ (Fig. 12). The result shows that the correlations for most sites, especially in eastern and southern Asia, pass the significant level of 0.1, though the correlation is not significant for part stattions in central Asia, western Asia, the coastal area in southeast Asia and western Russia. For example, in the 6-sites (Haerbin, Beijing, Qingdao, Shanghai, Yichang and Shantou) evenly distributed across eastern China (Fig. 12), all the correlations between reconstruction and observation pass the 0.01 significant level (Fig. 13). Moreover, the reconstructions could reproduce the most of extreme years, e.g., 1853, 1871, 1890, 1893, 1920 and 1921 in Beijing, 1875, 1876, 1889, 1891, 1892, 1921, 1929, 1931 and 1934 in Shanghai, and 1889, 1897, 1900, 1902, 1920, 1928, 1935 and 1937 in Yichang (Fig. 13). While the high $p$-values (i.e. $p>0.1$) of the positive correlation or negative correlation in part of stations (e.g., in India, southeast Asia, etc.) might be induced by both uncertainties from reconstructions based on few proxies available and observations in early times, because the instrumental data from these station usually extend to the 1880s and before (e.g., several stations in India extend to 1813) with missing records and using defective rain gauge in early time frequently. This assessment indicates that our reconstruction has high quality to show the precipitation variability in most of the study areas except for few grids in western Russia, the coastal area of southeast Asia and northern Japan. Thus, these datasets could be used to further study the spatiotemporal variability and underlying mechanisms of Asian precipitation since the pre-industrial era that is critically needed for climate modeling, prediction, and attribution.

For example, we use the dataset of Nov-Oct SPI Version B to investigate the spatiotemporal pattern of hydroclimate variability over Asia for 1700-2000 by Empirical Orthogonal Function (EOF) analysis. Figure 14a shows the first ten eigenvalues and their 95% confidence uncertainty intervals generated by the method from North et al. (1982). We find that only the first leading eigenvalue is independent, while the uncertainty intervals of other eigenvalues are overlapped and the cumulative explained variance of the first ten eigenvalues only accounts for 30.83% of the total (Fig. 14b). Such results indicate that there exists multiple spatial patterns of precipitation in Asia. Here we show some major characteristics for the first four modes, including spatial patterns (Fig. 14c-f), temporal changes (Fig. 15), and their correlations with winter sea surface temperature anomaly (SSTA) after high-pass filter (Fig. 16). Noted that the gridded SSTA data are from ERSSTv5 over 1854-2000 (Huang et al., 2017).

The spatial pattern of EOF1 has strong positive loadings over central Russia and a broad region from western Asia to central Asia to western China, while dominant negative loadings over monsoon region and eastern Russia (Fig.14c). The time-series for EOF1 has powerful inter-annual fluctuations over the considered period and significant decadal fluctuations in 1770-1820 (Fig.15a). Its high-frequency change (10-year high-pass filter) shows striking negative correlation with winter SSTA in central equatorial Pacific and Indian Ocean, but positive correlation in western tropical Pacific (Fig.16a), which is suggested that this mode is strongly affected by coupling oscillation in tropical oceans, i.e. Indo-Pacific tripole (Lian et al., 2013).

The EOF2 shows teleconnected pattern with negative loadings over western Asia, India, northern China, western and eastern Russia, but positive loadings in the rest regions (Fig.14d). The energy bands of the time-series for EOF2 are similar to those for EOF1 while its decadal fluctuations are expressed in 1840-1900 (Fig.15b). The inter-annual fluctuation of this

mode is significantly correlated with winter SSTA only in eastern tropical Pacific (Fig.16b), which indicate that EOF2 is dominated by ENSO.

The EOF3 and EOF4 both show multi-pole spatial pattern (Fig.14e-f) and their time series are dominated by decadal to multi-decadal scale fluctuations (Fig.15c-d). They express some significant decadal precipitation patterns in specific regions. For instance, precipitation over eastern China has two major patterns of decadal variation (Zheng et al., 2016), one is a dipole pattern divided by the Huai River and it is consistent with EOF3 in our study (Fig.16e), the other is a four-zone pattern (centered at South China, the Yangtze River Valley, North China Plain and Northeast China) which is similar to EOF4 (Fig.15f). In contrast, the high frequency fluctuations are relatively weak for these two EOFs and their links with SSTA are not significant.

### 3.3 Effectiveness of GLDD and Uncertainty

Limited by the spatial coverage and uneven distribution of available proxies, to reconstruct the grid dataset on past climate for large scale such as continental, hemispherical or global, it's necessary to search the proxy for calibration from a large area (so called "searching region" usually). In previous studies on past temperature (Christiansen and Ljungqvist, 2017) or hydro-climate reconstruction (Cook et al., 2010a; Shi et al., 2018; Shi et al., 2017), the searching region for each grid is set as a circular area with the same isotropic searching radius (ISR). However, as pointed out by Christiansen and Ljungqvist (2017) from the investigation on spatial decorrelation length in the Northern Hemispheric temperature filed, the searching radius for different target grid varies from less than 1000 km to more than 6000 km relying on target grid location and the searching direction along with the spatial pattern of coherence on temperature variation at different time scales. Since the spatial heterogeneity on precipitation variation is more evident than that on temperature variation, the proxy for a target hydroclimate reconstruction should be more sensitive to location and the searching direction.

Compared with searching proxies using isotropic searching radius, GLDD searches proxies for each target grid from the surrounding region of all connected grids where the robust coherence of precipitation variability and rainfall regime are coherent with the target grid. As shown in Fig. 17, there are evident difference in the maximum (Fig. 17a) and minimum distances (Fig. 17b) from the boundary of the searching region to each target grid across Asia. The maximum distance is 1000~2000 km for most grids in China, Mongolia, and central and northwest Russia, 2000~3000 km for most grids in India, central Asia, and southwest and eastern Russia, 3000~4000 km in Arabian Peninsula, and more than 4000 km for tropical islands. However, the minimum distance is only 250~750 km for most grids of study area, except very few grids in tropics. The difference between maximum and minimum distances (Fig. 17c) could reach 2000 km or more in regions with high topographic complexity, which means that the searching region is always in an irregular shape. Thus, for the area (e.g. the Tibetan Plateau and surrounding area) with complicated topography and multiplex hydroclimate variation, GLDD could identify the unique searching region (including shape and size) rigorously for each target grid. While for the area with homogeneous hydroclimate variability and rainfall regime, GLDD could capture the proxies far from the target grid, which could reconstruct well at the areas where proxy data are not present, such as the east of the Caspian Sea. Therefore, GLDD

could search the optimal proxies for hydroclimate reconstruction for each grid, and consequently improve the quality of reconstructed dataset (See the overall improvement of $R^2a$ compared with prior studies in section 3.2). For example, the $R^2a$ of our reconstruction for the grid of 90.0-92.5°E and 27.5-30.0°N in Fig. 5 reach 40.8%, which is 10.9% higher than that of Shi et al (2018) reconstruction by merging tree-ring and documentary records together via ISR.

    Yet, this study still exists the limitation and uncertainty. First, GLDD could only search the candidate proxies for the

reconstruction in a target grid from the surrounding region of all connected grids where the precipitation variability and rainfall regime are robust coherent with the target grid, and exclude the proxies from a tele-connected pattern. Thus, GLDD could result that the candidate proxies for the target grids are usually limited comparing to ISR approach. This might lead to not only enlarge the uncertainty but also induce missing data in the reconstruction for an area with large spatial heterogeneity on precipitation variation and rainfall regime due to very few candidate proxies available, such as central Russia and Arabian

Peninsula. Second, for tree-ring proxies, we use the same standardization method to build chronologies when raw measurements are available. However, about 4.5% of the tree-ring proxies do not have raw measurement file thus we have to use the processed chronologies with various standardization methods by different data providers. As the test for some sites, the difference between chronologies could reach 20% in maximum from different standardization methods (Li et al., 2011). This may also induce the uncertainty in the reconstruction. Thirdly, for documentary proxies, DW120 may use instrumental

precipitation data to identify the dryness/wetness grades since 1951, especially after 1979, which might lead to overestimation of the calibration and verification metrics in eastern China. Fortunately, the data of dryness/wetness grades before 1950 are completely derived from historical documents (Wang and Zhao, 1979). Thus, comparison between the Fig. 12-13 and Figs 6-7 by each site-grid could help us to assess the overestimation, and the result shows that the overestimation of $R^2a$ in this reconstruction is about 10% on average over eastern China.

**4 Data availability**

    The dataset (Liu et al., 2022b) can be accessed from https://doi.org/10.57760/sciencedb.01829. This dataset is licensed under a CC BY-SA 4.0 license.

    **5 Conclusions**

    In this study, we use a multi-proxy (mainly from tree-ring and historical documents with clear annual dating) network

containing 2912 series to reconstruct SPI for the wet season (Nov-Apr for west Asia and May-Oct for the others ) and annual (Nov-Oct) time scale since 1700 over Asia with the spatial resolution of 2.5° × 2.5°. Compare to the previous studies (Cook et al., 2010a; Feng et al., 2013; Shi et al., 2018; Shi et al., 2017), our reconstruction is conducted at the grid level by an improved calibration method, which could search proxies for a target grid by a new approach of GLDD from its connected areas within a sub-region having homogeneous rainfall regime and similar precipitation variability. Meanwhile, many new

proxies were used, mainly including additional 113 tree-ring width chronologies in the monsoon Asia and more than 6100 dry/wet grade data (23.7%) from historical documents in China. These additional proxies evidently improve the coverage and distribution of proxies, and their temporal homogeneity due to the reconstructed period being limited to 300 years only. This dataset is the first SPI reconstruction covering entire Asia based on pure proxies (without long-term observations or climate model constraints) and can be used to more clearly investigate the Asian precipitation change since 1700, and to test the paleoclimate simulation in the industrial period.

**Author contributions**

QG and JZ designed the study and planned the reconstructions. YL processed the experimental data, performed the computations and drafted the manuscript. JZ critically revised the manuscript. All authors discussed and contributed to the reconstruction and manuscript.

**Competing interests**

The authors declare that they have no conflict of interest.

**Acknowledgements**

This work was supported by the National Key R&D Program of China on Global change (2017YFA0603300) and the National Natural Science Foundation of China (42005043, 42175058). We acknowledge the World Data Center for Paleoclimatology (WDC-P, https://www.ncei.noaa.gov/products/paleoclimatology) for hosting and providing paleoclimatology database.

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

585

**Figure captions**

**Figure 1:** The study area and the spatial difference of rainfall regime with the wettest bimester (two consecutive months) shown by the monthly GPCP precipitation data from 1948-2019. The dot marker indicates that the grid lacks a clear wet season. Annual (Nov-Oct) SPI is reconstructed for all non-grey grids, while wet season (Nov-Apr and May-Oct) SPI is reconstructed in regions with black and brown boundaries respectively.

**Figure 2:** The flow-chart of SPI reconstruction over Asia for 1700-2000 in this study

**Figure 3:** Spatial (a) and temporal (b) distribution of proxies.

**Figure 4:** Proportion of available data for DW120 in original version (a) and after updating (b). Sites with a cross marker in (b) are excluded in reconstruction.

**Figure 5:** Demonstration of a grid SPI reconstruction for showing proxy-selecting by the GLDD approach. (a) The target grid (yellow square) and regions (light blue) that have significantly (at least $p<0.05$) positive correlated SPI change. (b) The searching region connected with the target grid and proxies in it. A proxy marker with a black edge means it is significantly (at least p<0.1) correlated with SPI change in the target grid. (c) Temporal coverage of picked proxy series and derived four segments based on available proxies. Proxies are listed in ascending order of $p$-value from bottom to top. When a segment has more than 5 proxies, the bottom 5 (solid patch) are used in BSR and the others (cross patch) are excluded. Proxies remain in the final BSR model are marked with plus signs. (d) Reconstructed SPI series and calibration $R^2a$ for each segment.

**Figure 6:** $R^2a$, RE, and CE for Nov-Oct SPI reconstruction by multi-proxies without using tree-ring density chronologies and width chronologies with negative correlations to precipitation.

**Figure 7:** $R^2a$, RE, and CE for Nov-Oct SPI reconstruction by multi-proxies including tree-ring density chronologies and width chronologies with negative correlations to precipitation.

**Figure 8:** $R^2a$, RE, and CE for wet season SPI reconstruction by multi-proxies without using tree-ring density chronologies and width chronologies with negative correlations to precipitation, the black line indicates the boundary of the region in which the wet season is Nov-Apr as that in Fig. 1.

**Figure 9:** $R^2a$, RE, and CE for wet season SPI reconstruction by multi-proxies including tree-ring density chronologies and width chronologies with negative correlations to precipitation, the black line indicates the boundary of the region in which the wet season is Nov-Apr as that in Fig. 1.

**Figure 10:** The maps of correlation between the wet season SPI reconstruction Version A of this study and four reconstructions in monsoon Asia by previous studies of Shi et al. (2018) (a), Shi et al. (2017) (b), Feng et al (2013) (c) and (Cook et al, 2010a) (d) respectively. Correlation values significant at 99% confidence are shown by dot marker.

**Figure 11:** Same as Figure 10 but for the wet season SPI reconstruction Version B.

**Figure 12:** Correlations between Nov-Oct precipitation anomaly percentage for weather stations with at least 30-year records before 1948 from GHCMm and Nov-Oct SPI reconstruction in corresponding grids. The six selected sites in eastern China are shown with black edges, and the comparisons between observation and reconstruction year by year in these sites will be shown as the examples in Figure 13.

**Figure 13:** Comparisons between Nov-Oct precipitation anomaly percentage for 6 sites across eastern China from Tao et al. (1997) and Nov-Oct SPI reconstruction in corresponding grids. (a) Haerbin (126.62°E, 45.68°N), (b) Beijing (116.28°E, 39.93°N), (c) Qingdao (120.33°E, 36.07°N), (d) Shanghai (121.43°E, 31.17°N), (e) Yichang (111.30°E, 30.70°N), and (f) Shantou (116.68°E, 23.40°N). Their locations are also shown in Fig. 12.

**Figure 14:** EOF analysis of Nov-Oct SPI reconstruction in Asia. (a) The first ten eigenvalues and their 95% uncertainty intervals. (b) The cumulative explained variance of the first ten eigenvalues. (c-f) Spatial patterns of EOF1-EOF4.

**Figure 15:** Temporal change and wavelet power spectrum of the time series (PC) for EOF1-EOF4 (a-d) shown in Fig. 14. PC is shown after normalization with a 10-year low-pass filter (black) applied to each. Spectral band significant above the 90% level are shown by black contours.

**Figure 16:** Field correlations between SSTA in winter and time series of EOF1 (a) and EOF2 (b) after 10 year high-pass filter. Correlation values significant at 95% confidence are shown by dot marker.

**Figure 17:** The maximum (a) and minimum (b) distance from boundary of the searching region to the target point, and their difference (c) for each grid.

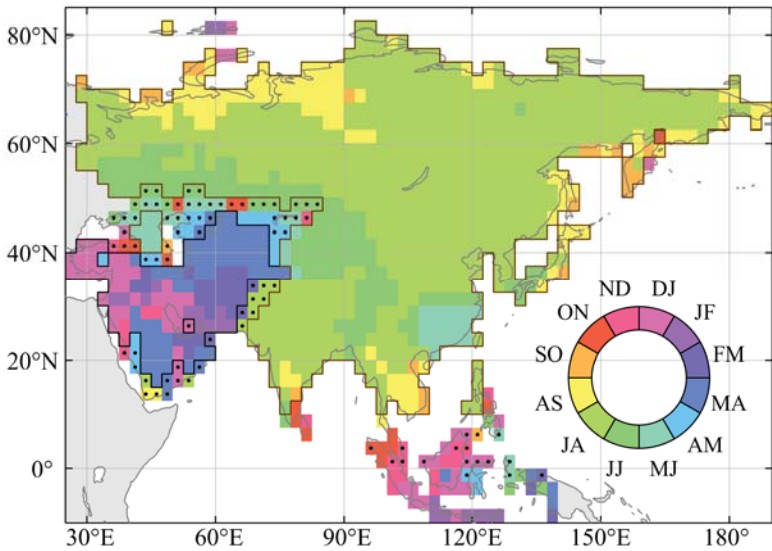

**Figure 1:** The study area and the spatial difference of rainfall regime with the wettest bimester (two consecutive months) shown by the monthly GPCP precipitation data from 1948-2019. The dot marker indicates that the grid lacks a clear wet season. Annual (Nov-Oct) SPI is reconstructed for all non-grey grids, while wet season (Nov-Apr and May-Oct) SPI is reconstructed in regions with black and brown boundaries respectively.

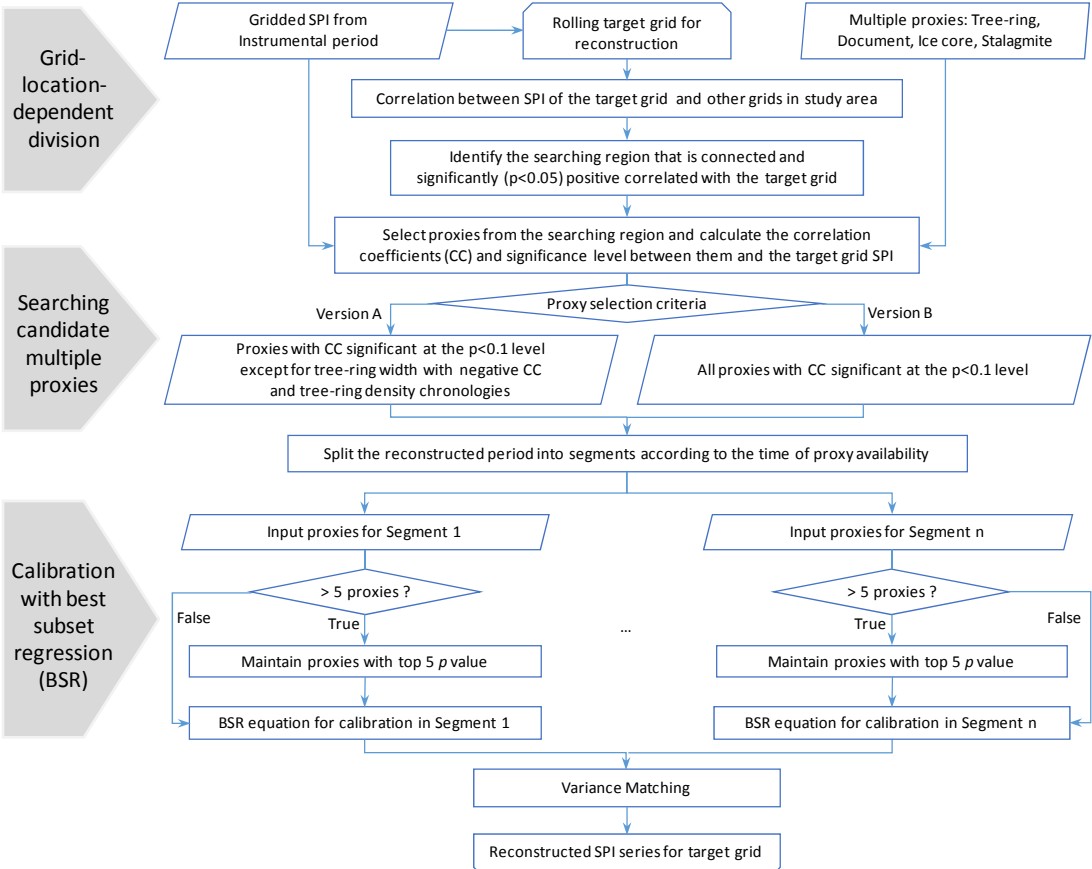

**Figure 2:** The flow-chart of SPI reconstruction over Asia for 1700-2000 in this study


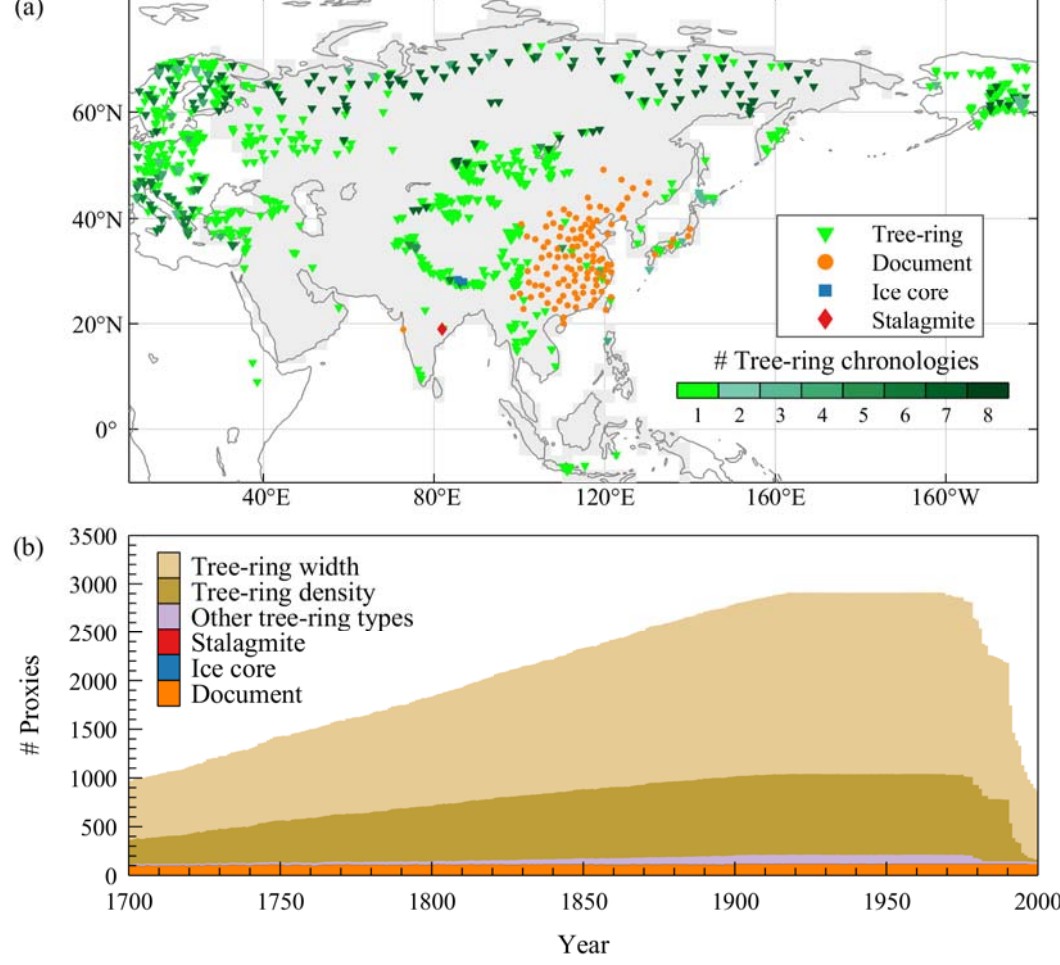

**Figure 3:** Spatial (a) and temporal (b) distribution of proxies.

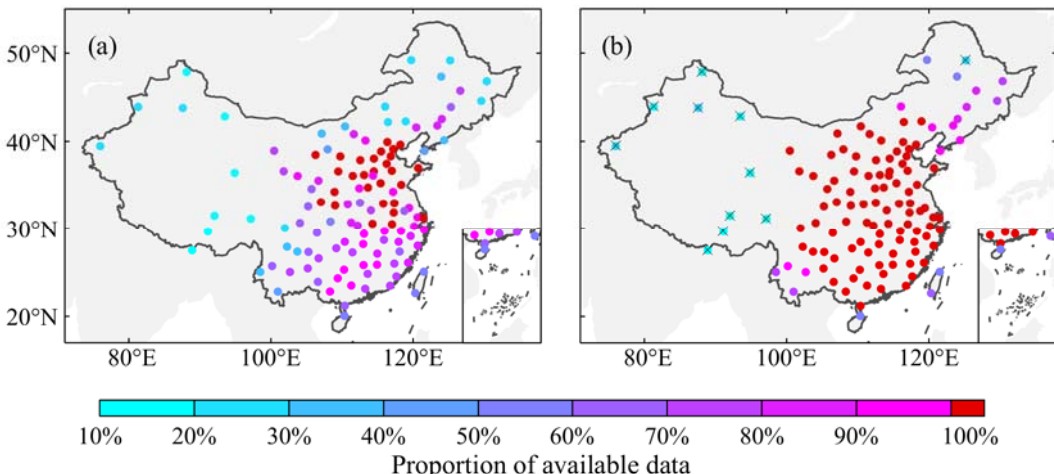


**Figure 4:** Proportion of available data for DW120 in original version (a) and after updating (b). Sites with a cross marker in (b) are excluded in reconstruction.

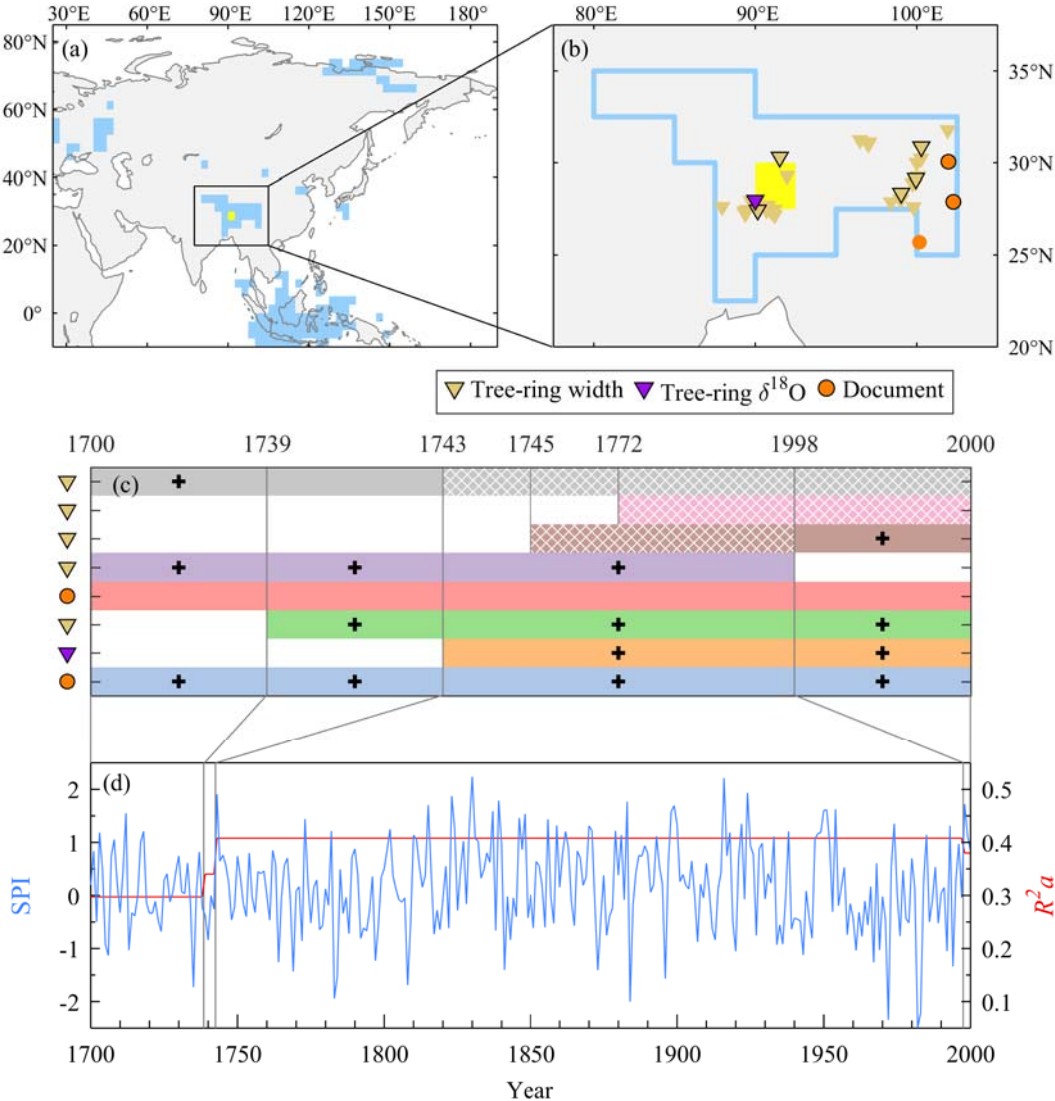


**Figure 5:** Demonstration of a grid SPI reconstruction for showing proxy-selecting by the GLDD approach. (a) The target grid (yellow square) and regions (light blue) that have significantly (at least *p*<0.05) positive correlated SPI change. (b) The searching region connected with the target grid and proxies in it. A proxy marker with a black edge means it is significantly (at least p<0.1) correlated with SPI change in the target grid. (c) Temporal coverage of picked proxy series and derived four

segments based on available proxies. Proxies are listed in ascending order of *p*-value from bottom to top. When a segment has more than 5 proxies, the bottom 5 (solid patch) are used in BSR and the others (cross patch) are excluded. Proxies remain in the final BSR model are marked with plus signs. (d) Reconstructed SPI series and calibration $R^2a$ for each segment.

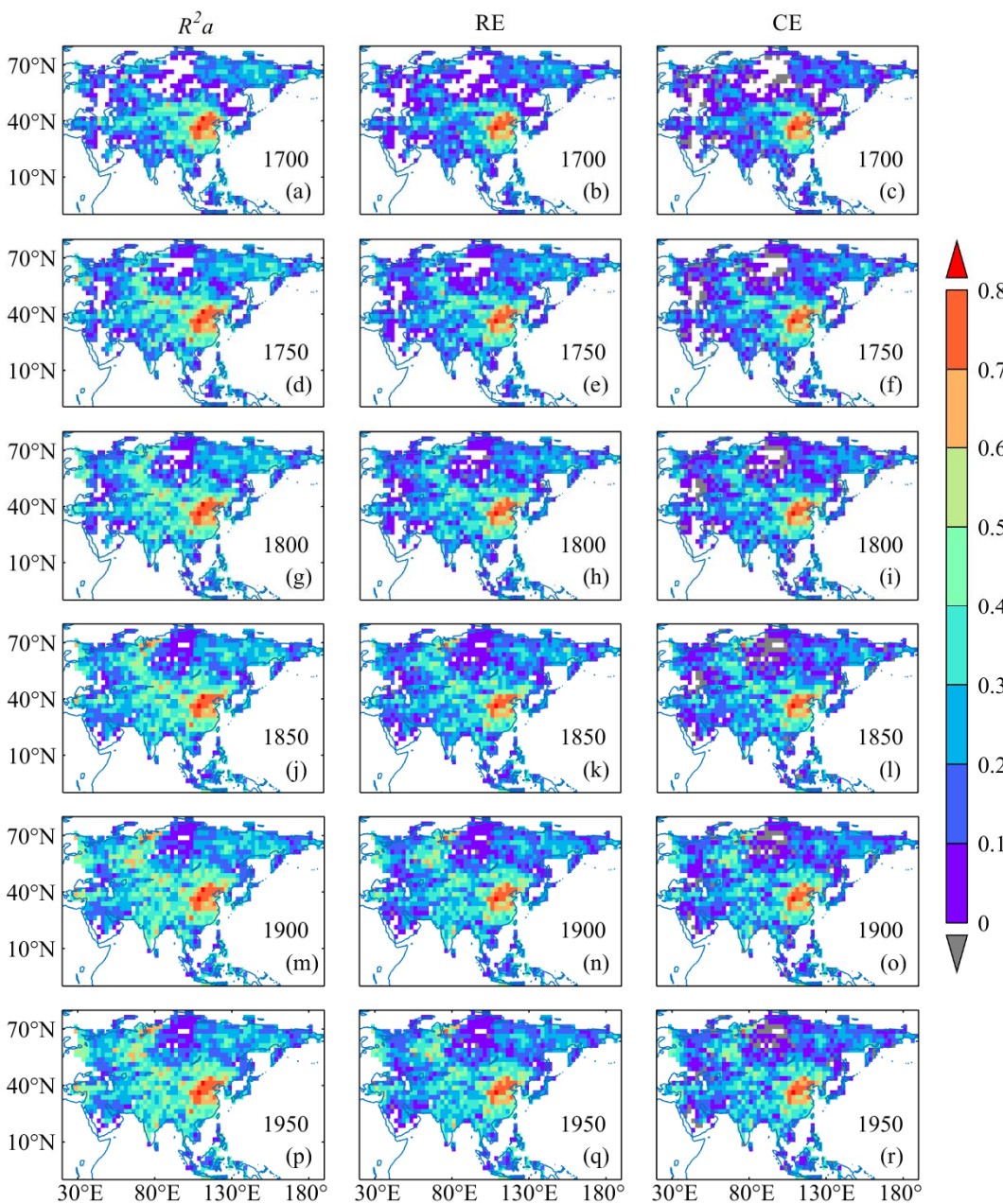

**Figure 6:** $R^2a$, RE, and CE for Nov-Oct SPI reconstruction by multi-proxies without using tree-ring density chronologies and width chronologies with negative correlations to precipitation.

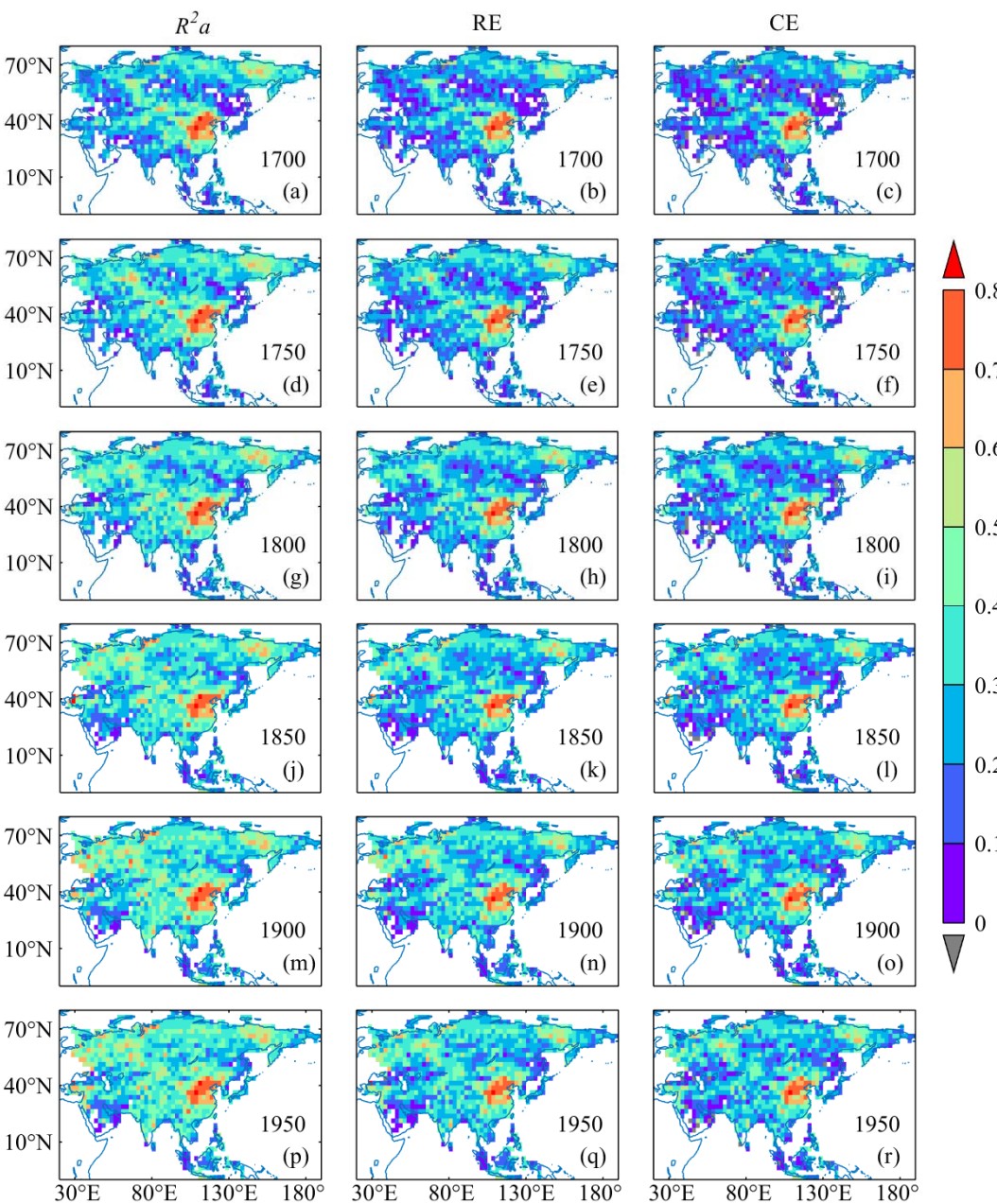

**Figure 7:** $R^2a$, RE, and CE for Nov-Oct SPI reconstruction by multi-proxies including tree-ring density chronologies and
width chronologies with negative correlations to precipitation.


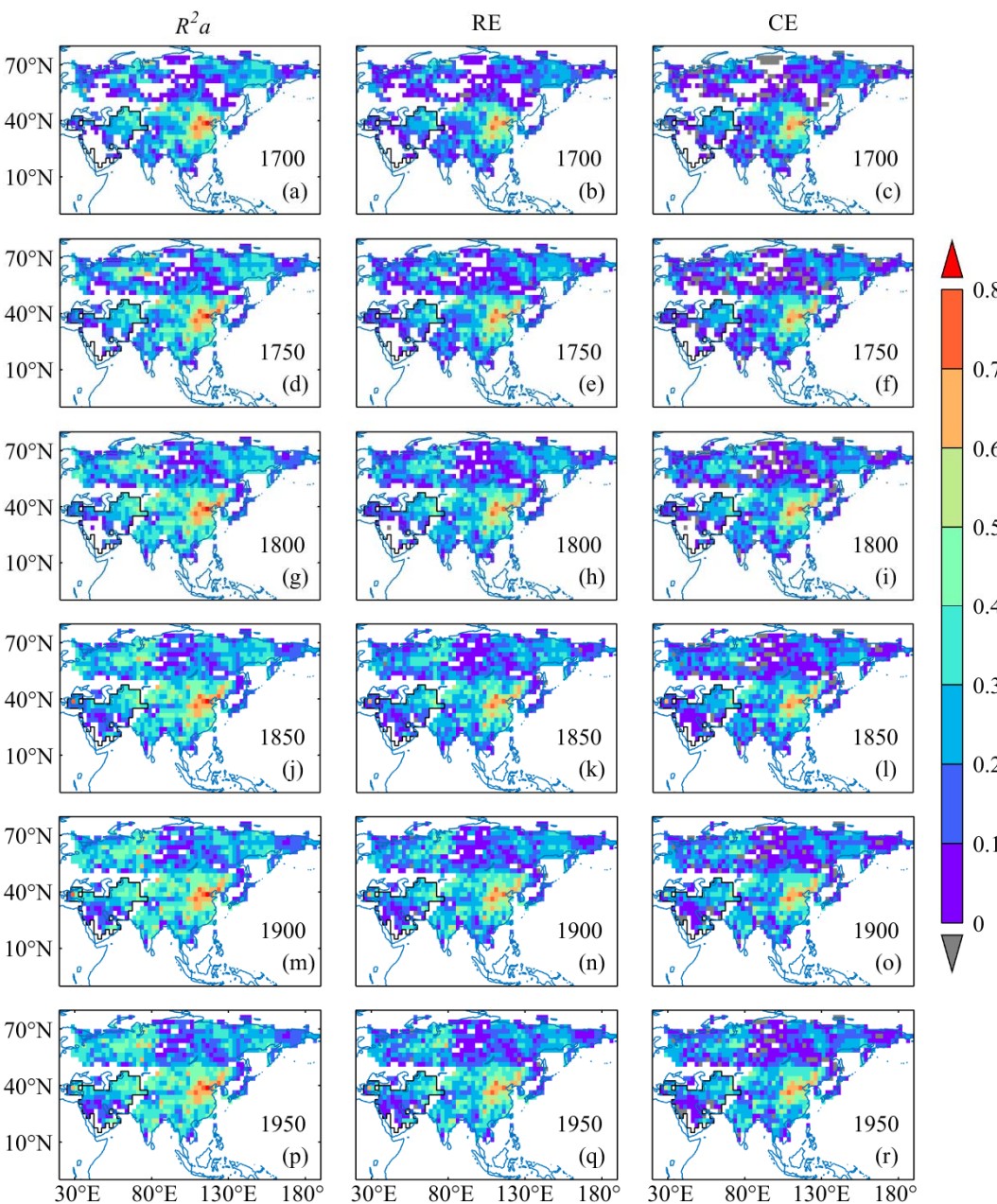

**Figure 8:** $R^2a$, RE, and CE for wet season SPI reconstruction by multi-proxies without using tree-ring density chronologies and width chronologies with negative correlations to precipitation, the black line indicates the boundary of the region in which the wet season is Nov-Apr as that in Fig. 1.

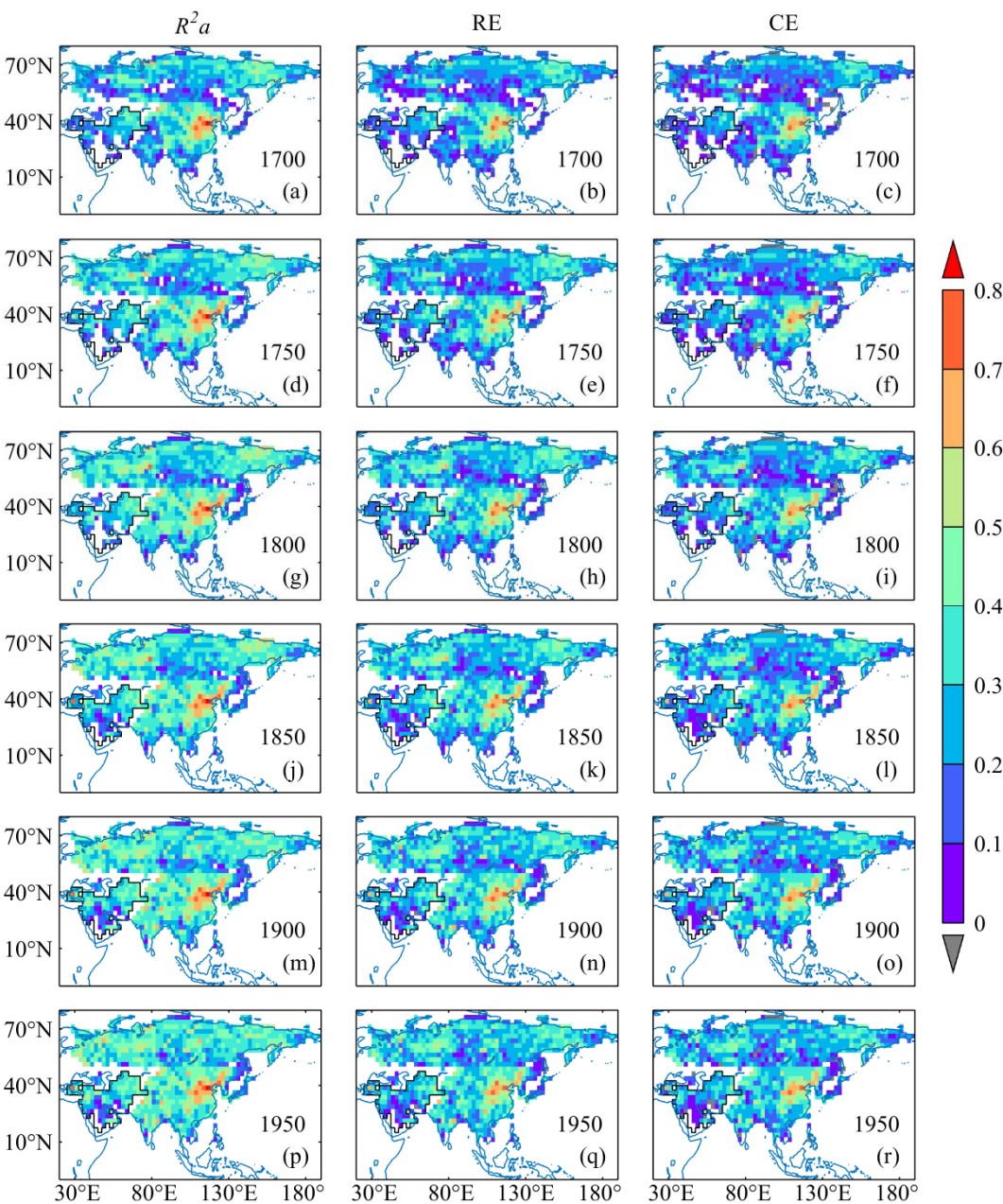

**Figure 9:** $R^2a$, RE, and CE for wet season SPI reconstruction by multi-proxies including tree-ring density chronologies and width chronologies with negative correlations to precipitation, the black line indicates the boundary of the region in which
the wet season is Nov-Apr as that in Fig. 1.

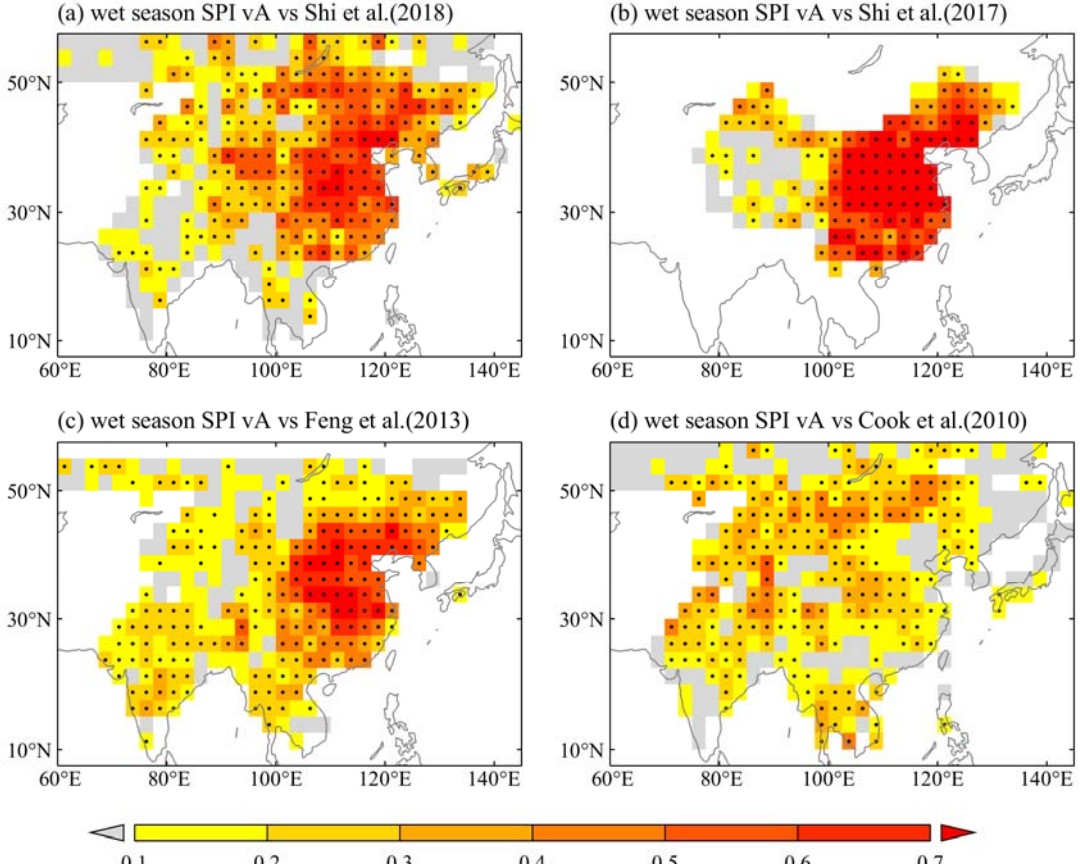

**Figure 10:** The maps of correlation between the wet season SPI reconstruction Version A of this study and four reconstructions in monsoon Asia by previous studies of Shi et al. (2018) (a), Shi et al. (2017) (b), Feng et al (2013) (c) and 700 (Cook et al, 2010a) (d) respectively. Correlation values significant at 99% confidence are shown by dot marker.

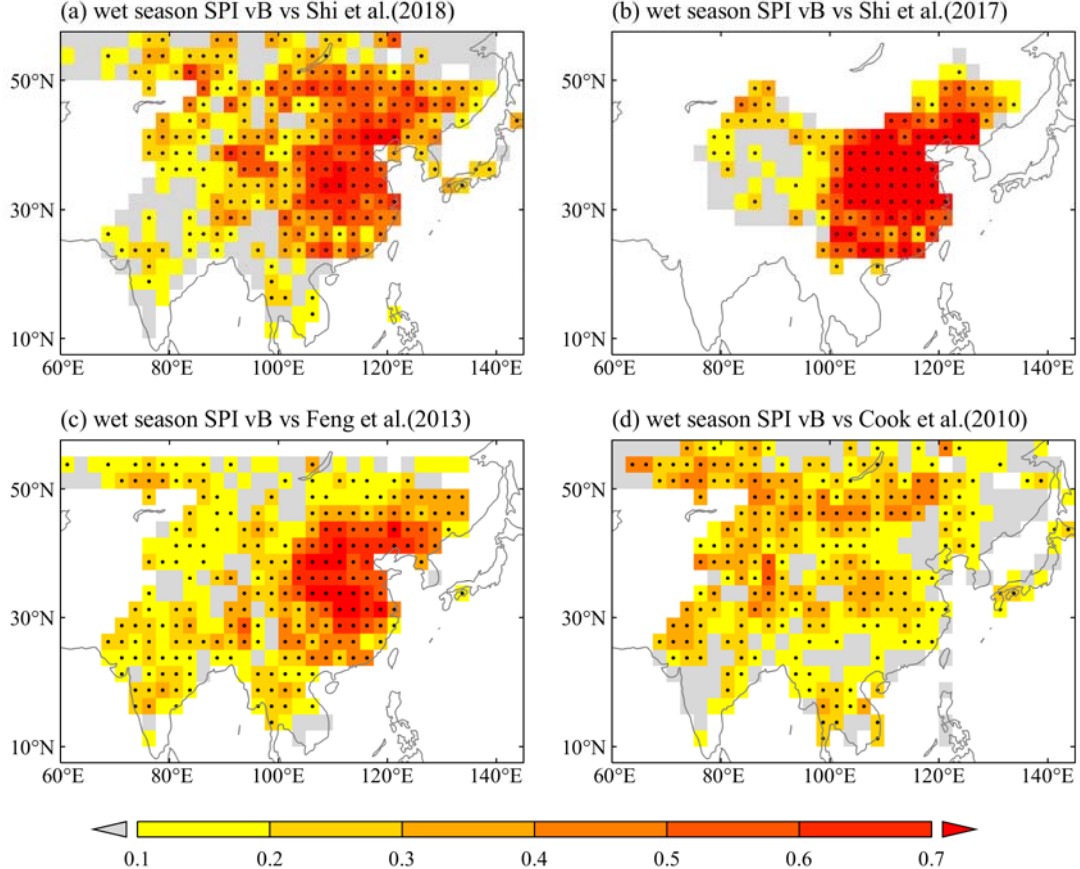

**Figure 11:** Same as Figure 10 but for the wet season SPI reconstruction Version B.

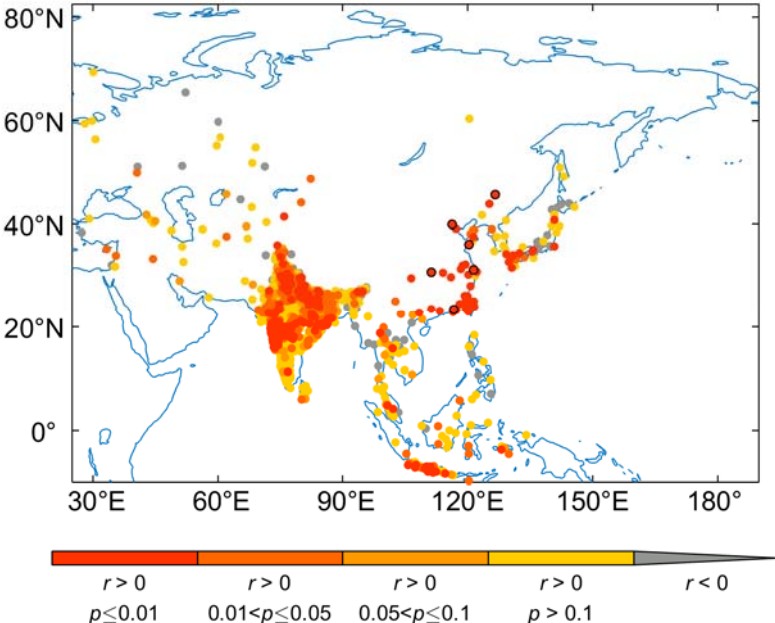

**Figure 12:** Correlations between Nov-Oct precipitation anomaly percentage for weather stations with at least 30-year records before 1948 from GHCMm and Nov-Oct SPI reconstruction in corresponding grids. The six selected sites in eastern China are shown with black edges, and the comparisons between observation and reconstruction year by year in these sites will be shown as the examples in Fig. 13.


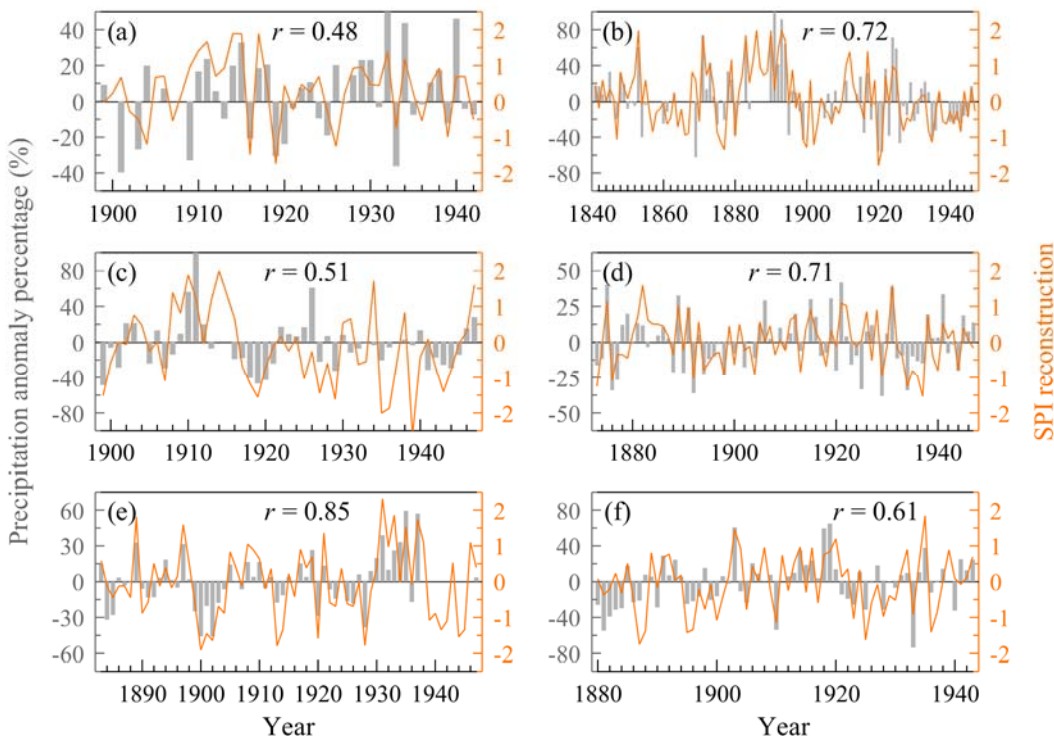

**Figure 13:** Comparisons between Nov-Oct precipitation anomaly percentage for 6 sites across eastern China from Tao et al. (1997) and Nov-Oct SPI reconstruction in corresponding grids. (a) Haerbin (126.62°E, 45.68°N), (b) Beijing (116.28°E, 39.93°N), (c) Qingdao (120.33°E, 36.07°N), (d) Shanghai (121.43°E, 31.17°N), (e) Yichang (111.30°E, 30.70°N), and (f) Shantou (116.68°E, 23.40°N). Their locations are also shown in Fig. 12.

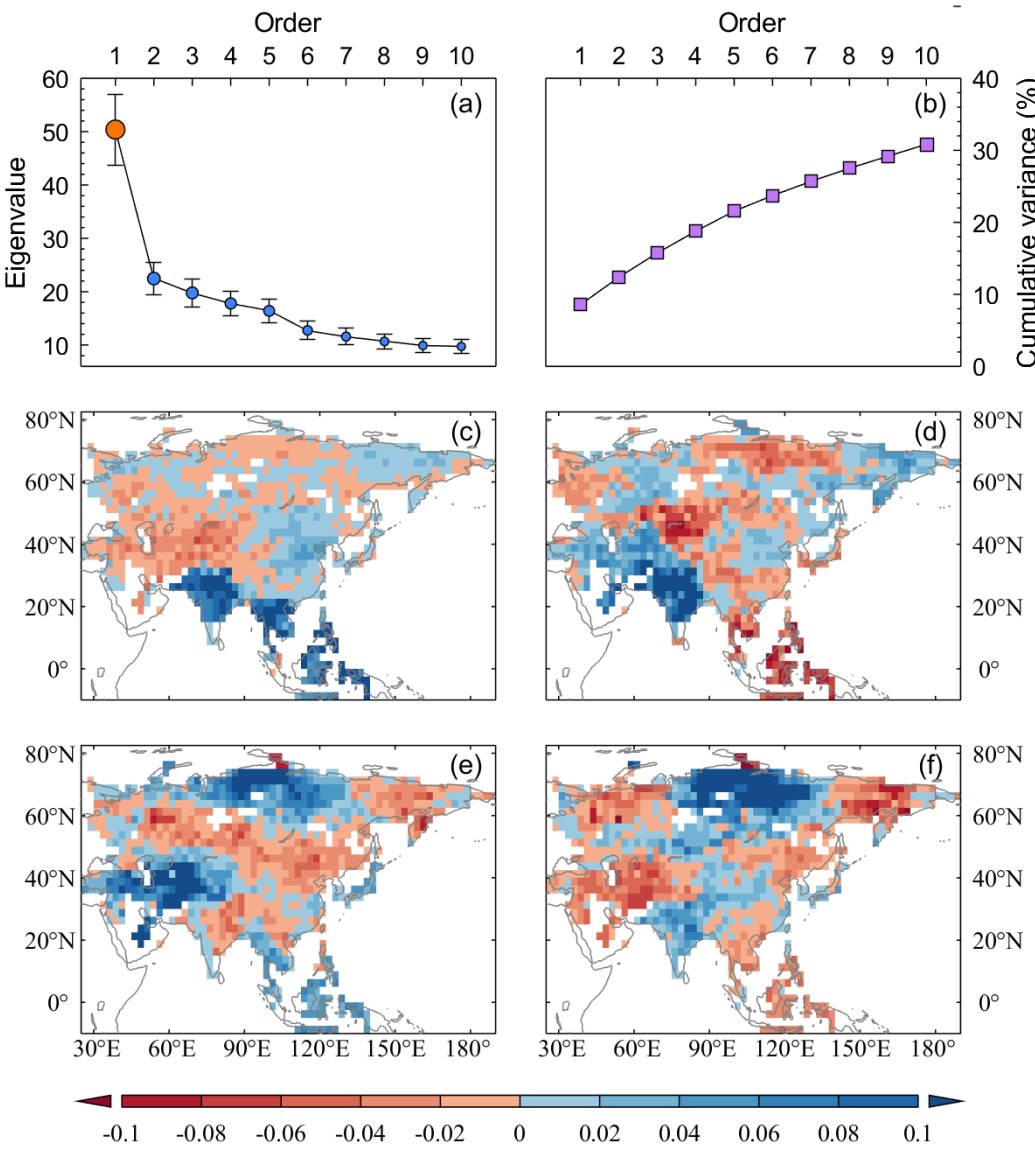

**Figure 14:** EOF analysis of Nov-Oct SPI reconstruction in Asia. (a) The first ten eigenvalues and their 95% uncertainty intervals. (b) The cumulative explained variance of the first ten eigenvalues. (c-f) Spatial patterns of EOF1-EOF4.

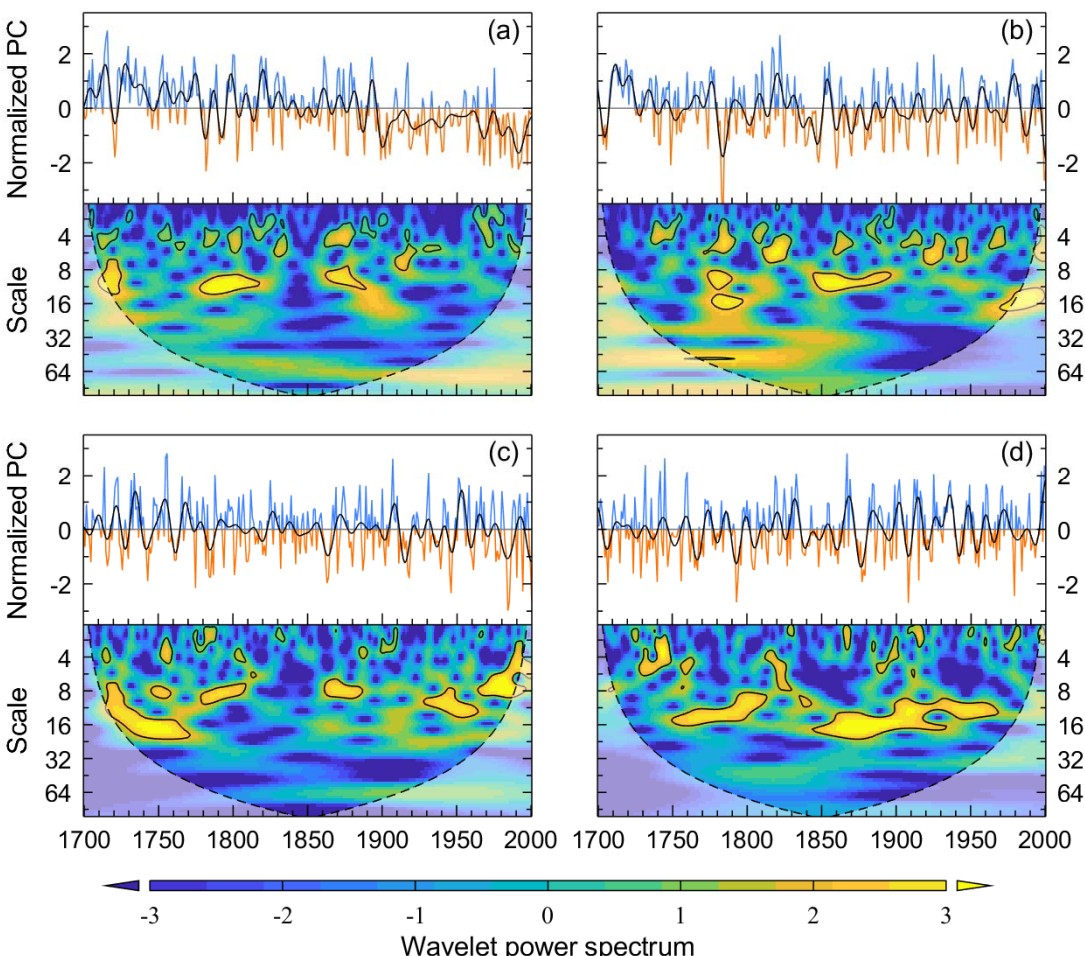

**Figure 15:** Temporal change and wavelet power spectrum of the time series (PC) for EOF1-EOF4 (a-d) shown in Fig. 13. PC is shown after normalization with a 10-year low-pass filter (black) applied to each. Spectral band significant above the 90% level are shown by black contours.

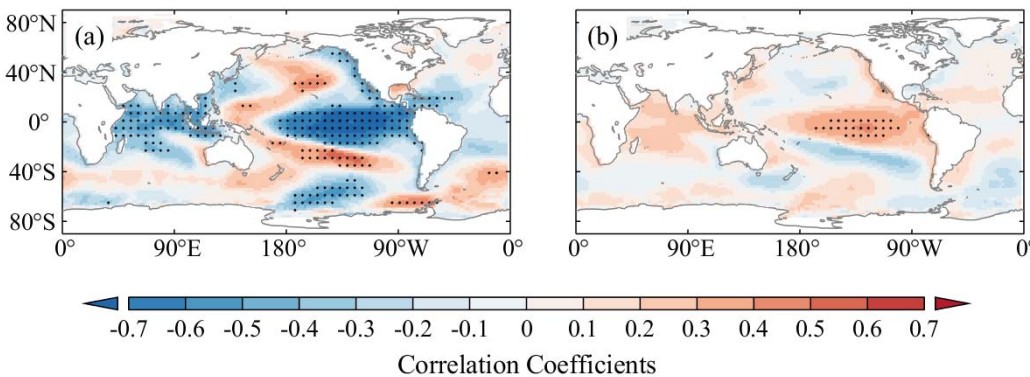

**Figure 16:** Field correlations between SSTA in winter and time series of EOF1 (a) and EOF2 (b) after 10 year high-pass filter. Correlation values significant at 95% confidence are shown by dot marker.

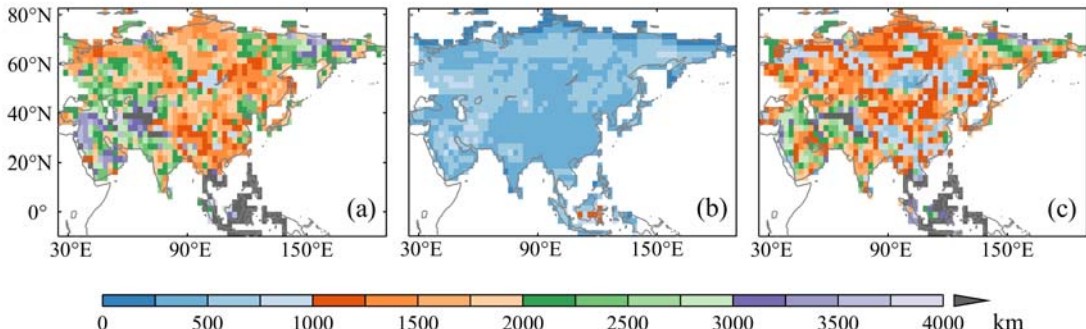

**Figure 17:** The maximum (a) and minimum (b) distance from boundary of the searching region to the target point, and their difference (c) for each grid.

735