# Peer review of "A dataset of standard precipitation index reconstructed from multiproxies over Asia for the past 300 years"

_Earth System Science Data, 2022_

## Author Comment (AC1)

**RC 1**

The manuscript titled "A dataset of standard precipitation index reconstructed from multiproxies over Asia for the past 300 years" by Liu et al presents a great effort of extending the spatial and temporal reconstruction of the hydroclimate variation in Asia, which would serve as a valuable component to the PAGES 2K family. I therefore recommend the publication of this work, after the authors addressing the following comments and questions:

**Major comments**

1.   The study area covers a larger region that what has been discussed in the introduction. Please provide a brief rationale of why choosing the study area, especially the Russian territory in Europe, and add in the introduction, the research progress has been made the areas not covered in the four gridded datasets.

   Accepted and revised. Flood and drought and their associated social and economic damages are often reported and investigated at the national level. For example, all over the world, seven of the top ten countries with the largest number of population affected by climate-related disasters (mainly flood and drought) are located in Asia (CRED and UNISDR, 2015). Since most of Russian territory is located in Asia, to keep the data integrity at the national level, the whole Russian territory is included in this study. We also add the western and northern regions of Asia that were not covered in the prior four gridded datasets. The following sentences are added:

   (1) At the national level, seven of the top ten countries in the world with the largest number of population affected by climate-related disasters (mainly flood and drought) are located in Asia (CRED and UNISDR, 2015). (Line 23-24)

   (2) Noted that most of Russian territory is located in Asia, to keep the data integrity at the national level, the whole Russian territory is included in this study. (Line 63-64)

2.   One novelty of this reconstruction is the new approach of the grid-location-dependent division (GLDD), but the description and validation of the method, in my opinion, is not sufficient. For example, if GLDD is original to this study, then it deserves a separate subsection and should be developed from a theoretical framework; or if it is a new application of an existing method,

then the original reference(s) should be given.

Accepted and revised. The GLDD is a new approach developed by this study to identify a region with robust coherent SPI variation surrounding the target grid for optimal selecting proxies in the target grid SPI reconstruction. We rewrote the section 2.4 and added the statements on how to develop GLDD, including its fundament, aim and arithmetic as following:

This approach is developed according to the regional division of the coherence of inter-annual precipitation variations in context of the spatial pattern of rainfall regime, to ensure the proxies in the searching region can well indicate SPI variability in the target grid. We divided the regions from the spatial pattern of correlation coefficient (CC) between SPI of each target grid for SPI reconstruction and the other grids within study area, which is calculated from instrumental SPI data. The searching region is defined as all connected grids surrounding the target grid with CC passing 0.05 significant level. Thus, this searching region has the robust coherence of precipitation variability and rainfall regime with the target grid, and the proxies in this region have best spatial representativeness to target grid. Since this regional division is dependent on the grid-location by rolling the target grid, this approach was called "grid-location-dependent division (GLDD)" in our study (Line 170-178).

3. A work-flow-chart illustrating the steps taken to reconstruction the index will be very helpful, as multi-proxy datasets have been applied during the procedure, and different data sites have been involved and some eliminated at various stages.

Accepted and revised. We add a work-flow-chart (Fig. 2) to illustrate the reconstruction procedures in section 2.1 (Line 73) with the following Figure.

[Figure]

Fig. 2. The flow-chart of SPI reconstruction over Asia for 1700-2000 in this study

4. Along with comment (3), it would be nice to show in Figure 1 where the "135 ring-width chronologies" are and what time frame do they cover. It would also be helpful to make a table listing the number of proxies (sites) used for construction of Nov-Oct SPI2 Version A and B, and wet season Version A and B, respectively, at different time intervals.

Accepted and revised. We add a table for all proxies used in this study (including their location, time coverage and original source) to help users comparing our proxies with any other public dataset. Unfortunately, we are not able to show the detailed information of the additional 135 chronologies in the Fig.1 because prior papers only depicted that they use a total of 453 tree-ring chronologies in the Monsoon Asia region, but didn't include the location information or other metadata for those chronologies.

Noted that we exclude the short proxies with less than 30 records before 1948 for reconstruction due to the change of cross-validation method suggested by reviewer 2, thus the total number of tree-ring chronologies used in this study is only 2772 now and only increase 113 ring-width chronologies compare to that of 453. We made this change in Line 101-106 and Line

125.

We also add a variable nPrx in all datasets (Nov-Oct SPI2 Version A and B, and wet season Version A and B) to present the number of proxies used for each grid at different time intervals.

5. The merge of DW65 into DW120 (Line 132-139) needs to be illustrated more clearly. For example, how were the 2045 records of DW65 added? Is there a calibration or verification performed on both datasets? How were the 2045 added records fit to the isolines? Please also provide the rationale for including the 4 ice cores and 1 stalagmite record, and the important data information.

Accepted and revised. We rewrote the sentences to clarify how we use DW63 (DW65 is a typo in original manuscript) to replace DW120 as following:

We added missing records in DW120 by two steps. The first is adding the missing data in DW120 from another dryness/wetness grade dataset for 63 sites in central eastern China (DW63) developed by Zhang (1996). Noted that all of sites of DW63 are included in DW120 and the grading criteria for DW63 is same as DW120. Since DW63 was reconstructed from more abundant historical documents (such as the drought/flood descriptions recorded in the memoirs and archives of the Qing Dynasty), it had fewer missing records with 100% data availability after 1700. Therefore, all missing records of DW120 in central eastern China are added from DW63, which supplements 2045 grade records in total. The second is interpolation from the isoline map of DW120 for individual years when most sites have available data (CAMS, 1981), which supplements 4121 grade records. Since DW63 was reconstructed by the same grading criteria as DW120, both of the added 2045 grade records from DW63 and the added 4121 records from the yearly isoline map of DW120 match with the original available data. (Line 144-153)

The 4 ice cores in the Himalayas and one stalagmite in India used in our work have been proved to indicate hydro-climatic change by previous studies (Thompson et al., 2000; Sinha et al., 2011; Qin et al., 2002) and they are annual-resolved proxies. We do not use other ice cores and stalagmites in WDC-P because they have lower temporal resolutions. The information of their location, time coverage and original source are provided in metadata file.

6. The "Results and Discussion" section could be enriched by illustrating some examples of its

usability. For example, the new reconstruction could benefit the study of Asian monsoon responses to ENSO and volcanic eruptions (Liu et al., 2022). Or it could be utilized to study the interannual to centennial spatiotemporal variability in hydroclimate reconstruction in Asia (therefore also echoing one of the motivations of this study suggested in Line 22-24).

Accepted and revised. We added the spatiotemporal pattern of hydroclimate variability over Asia for 1700-2000 by Empirical Orthogonal Function (EOF) analysis in section 3.2 (Line 301-307 with Fig 14-16 ) as following:

For example, we use the dataset of Nov-Oct SPI Version B to investigate the spatiotemporal pattern of hydroclimate variability over Asia for 1700-2000 by Empirical Orthogonal Function (EOF) analysis. Figure 14a shows the first ten eigenvalues and their 95% confidence uncertainty intervals generated by the method from North et al. (1982). We find that only the first leading eigenvalue is independent, while the uncertainty intervals of other eigenvalues are overlapped and the cumulative explained variance of the first ten eigenvalues only accounts for 30.83% of the total (Fig. 14b). Such results indicate that there exists multiple spatial patterns of precipitation in Asia. Here we show some major characteristics for the first four modes, including spatial patterns (Fig. 14c-f), temporal changes (Fig. 15), and their correlations with winter sea surface temperature anomaly (SSTA) after high-pass filter (Fig. 16). Noted that the gridded SSTA data are from ERSSTv5 over 1854-2000 (Huang et al., 2017).

The spatial pattern of EOF1 has strong positive loadings over central Russia and a broad region from western Asia to central Asia to western China, while dominant negative loadings over monsoon region and eastern Russia (Fig.14c). The time-series for EOF1 has powerful inter-annual fluctuations over the considered period and significant decadal fluctuations in 1770-1820 (Fig.15a). Its high-frequency change (10-year high-pass filter) shows striking negative correlation with winter SSTA in central equatorial Pacific and Indian Ocean, but positive correlation in western tropical Pacific (Fig.16a), which is suggested that this mode is strongly affected by coupling oscillation in tropical oceans, i.e. Indo-Pacific tripole (Lian et al., 2013).

The EOF2 shows teleconnected pattern with negative loadings over western Asia, India, northern China, western and eastern Russia, but positive loadings in the rest regions (Fig.14d). The energy bands of the time-series for EOF2 are similar to those for EOF1 while its decadal fluctuations are expressed in 1840-1900 (Fig.15b). The inter-annual fluctuation of this mode is significantly correlated with winter SSTA only in eastern tropical Pacific (Fig.16b), which indicate

that EOF2 is dominated by ENSO.

The EOF3 and EOF4 both show multi-pole spatial pattern (Fig.14e-f) and their time series are dominated by decadal to multi-decadal scale fluctuations (Fig.15c-d). They express some significant decadal precipitation patterns in specific regions. For instance, precipitation over eastern China has two major patterns of decadal variation (Zheng et al., 2016), one is a dipole pattern divided by the Huai River and it is consistent with EOF3 in our study (Fig.16e), the other is a four-zone pattern (centered at South China, the Yangtze River Valley, North China Plain and Northeast China) which is similar to EOF4 (Fig.15f). In contrast, the high frequency fluctuations are relatively weak for these two EOFs and their links with SSTA are not significant.

[Figure]

**Figure 14:** EOF analysis of Nov-Oct SPI reconstruction in Asia. (a) The first ten eigenvalues and their 95% uncertainty intervals. (b) The cumulative explained variance of the first ten eigenvalues. (c-f) Spatial patterns of EOF1-EOF4.

[Figure]

**Figure 15:** Temporal change and wavelet power spectrum of the time series (PC) for EOF1-EOF4 (a-d) shown in Fig. 13. PC is shown after normalization with a 10-year low-pass filter (black) applied to each. Spectral band significant above the 90% level are shown by black contours.

[Figure]

**Figure 16:** Field correlations between SSTA in winter and time series of EOF1 (a) and EOF2 (b) after 10 year high-pass filter. Correlation values significant at 95% confidence are shown by dot marker.

7. It is also important to discussion the uncertainties or limitations of this study, in terms of the

methodology or the changing of reconstruction quality along time or in some regions. For example, it would be nice to hear the authors' opinion on how is the GLDD method compare to the ones already used in PAGES 2K (Christiansen & Ljungqvist 2017).

Accepted and revised. We add the section 3.3 to discuss the effectiveness (including the merits and limitations) of GLDD and Uncertainty of this study is as followed (Line 330-373 with Fig 17).

**3.3 Effectiveness of GLDD and Uncertainty**

[revised manuscript text omitted]

Minor comments:

8.    Line 20-22, please add a couple of the most up-to-date studies, such as Wei et al (2020) on the topic.

Accepted and added.

9.    Line 53-54, please provide the references of the "many new proxies achieved in recent years" if they are from published sources.

Accepted and added.

10.  Line 55, please add "over the past 300 years" following "reconstruction effort on seasonal to annual precipitation variability", as a way to address the opening question (Line 22-24) of lacking interannual to centennial spatiotemporal variability.

Accepted and added.

11.  The subsection 2.2 on instrumental needs more details, for example, why does a separate GPCC precipitation data is used to identify the wet seasons of different regions in Asia? Some information about the different rainfall regimes and their driving mechanism will be helpful.

Accepted and added. Because moisture sources are different across Asia throughout the year and the wet season could be roughly classified as two terms of Nov-Apr and May-Oct, we use GPCC data to identify the spatial pattern of the wet season for SPI reconstruction in the 2.5º×2.5º gridded scale induced by different regional rainfall regimes (Line 86-92).

The mechanism for different rainfall regimes is complex and beyond the scope of this study, we cited some related references (Bombardi et al., 2019; Peng et al., 2020; Nieto et al., 2019) to address this point.

12. I believe there are some typo in some of the years mentioned in Line 173-178. For example, it should be "4 candidate proxies for 1700-1739 (Fig.4c)" instead of 1939. Please double check and correct all relevant typos.

Accepted and corrected.

---

## Author Comment (AC2)

**RC 2**

Dear authors of the manuscript "A dataset of standard precipitation index reconstructed from multi-proxies over Asia for the past 300 years", this is an interesting and fruitful work. The authors reconstructed the gridded precipitation dataset over Asia using the new method and the new dataset. I learned new information for the new method. I recommend this manuscript is accepted for publication after a minor revision.

Major comments:

1. It is suggested to compare this reconstruction with other four reconstructions. e.g., the correlation map.

Accepted and added (Line 268-278 with Fig 10 and 11) as follows:

In addition, the maps of correlation between our wet season SPI reconstructions and four reconstructions in monsoon Asia by previous studies show that most of grids pass 0.01 significant level (Fig. 10-11). Specifically, for the correlation (Fig. 10a, Fig. 11a) between our wet season SPI reconstruction Version A/B and the JJA precipitation reconstruction by Shi et al (2018), there are 63.2%/64.1% of all grids passed the 0.01 significant level, in which the value of correlation coefficients for central eastern China are almost higher than 0.60. Similar results are also found for the correlations between our reconstruction versus the May-September precipitation anomaly reconstruction by Shi et al (2017) in China (Fig. 10b, Fig. 11b) and the May-September precipitation reconstruction over monsoon Asia (Fig. 10c, Fig. 11c) by Feng et al (2013). Even for the correlations between our wet season SPI reconstruction versus JJA PDSI reconstruction for monsoon Asian (Fig. 10d, Fig. 11d) by Cook et al (2010a) only using tree-ring, 57.4% (for our reconstruction Version A versus JJA PDSI reconstruction) and 58.8% (for our reconstruction Version B versus JJA PDSI reconstruction) of all grids passed the 0.01 significant level.

[Figure]

**Figure 10:** The maps of correlation between the wet season SPI reconstruction Version A of this study and four reconstructions in monsoon Asia by previous studies of Shi et al. (2018) (a), Shi et al. (2017) (b), Feng et al (2013) (c) and (Cook et al, 2010a) (d) respectively. Correlation values significant at 99% confidence are shown by dot marker.

[Figure]

**Figure 11:** Same as Figure 10 but for the wet season SPI reconstruction Version B.

2. It is encouraged to archive the raw data and detailed information of all proxy records in the public database.

Accepted and revised. We added a table for all proxies used in this study (including their location, time coverage and original source) to help users comparing our proxies with any other public dataset.

Specific Comments:

1. Page 4, Line 121, replace 'Cma' with 'Chinese Academy of Meteorological Science'.

Accepted and revised.

2. Page 6, line 186, the leave-one-out cross validation is not a correct way to validate the regress

model for highly autocorrelated time series, e.g. the tree-ring width chronology. It is encouraged to use a split calibration-verification procedure.

Accepted and revised. We change the cross validation method by a state-of-the-art 4-fold rolling window cross-validation procedure (Nguyen et al., 2020). The related information in the paragraph is rewrote as following with all related data updated in the dataset.

Then, we use BSR to establish 4 calibration equations for SPI reconstruction in 1700-1738, 1739-1942, 1943-1997, 1998-2000 respectively, in which the best subset selection is determined by maximizing the Coefficient of Efficiency (CE) (Cook et al., 1994) calculated by a state-of-the-art 4-fold rolling window cross-validation procedure (Nguyen et al., 2020). Another commonly used validation parameter, reduction of error (RE), is also calculated from the same procedure. (Line 213-217)

---

## Author Comment (AC4)

**General comments**

This paper combines a vast amount of paleoclimate information consisting of different proxies such as tree rings and documentary materials to reconstruct annual and rainy season precipitation across Asia in a grid of 2.5 degrees over the past 300 years. Reconstructions of precipitation in such a wide area has been done several times in the past, but in this paper, not only new data that has not been reflected in previous studies has been added, but also a method called GLDD, which is a new method of sorting proxy data, is effectively used, and a very accurate reconstruction is realized. Therefore, the usefulness of the obtained data set is very large.

On the other hand, with regard to the evaluation of the results of this paper and future issues, several points that should be added or improved were recognized as follows.

1) GLDD is considered a very good proxy sorting method. In fact, this method can be used to reconstruct paleoclimate fairly accurately even in areas where proxy data is not present, such as the eastern part of the Caspian Sea. This means that, in some places, the "searching region" of the grid is very large. In order to help readers better understand the effectiveness of GLDD, it would be good to have a map showing the size of the "searching region" for each grid.

Accepted and revised. We add the maximum and minimum distances from the boundary of the searching region to each target grid in section 3.3 with Figure 12a-c as following (Line 341-352):

As shown in Fig. 17, there are evident difference in the maximum (Fig. 17a) and minimum distances (Fig. 17b) from the boundary of the searching region to each target grid across Asia. The maximum distance is 1000~2000 km for most grids in China, Mongolia, and central and northwest Russia, 2000~3000 km for most grids in India, central Asia, and southwest and eastern Russia, 3000~4000 km in Arabian Peninsula, and more than 4000 km for tropical islands. However, the minimum distance is only 250~750 km for most grids of study area, except very few grids in tropics. The difference between maximum and minimum distances (Fig. 17c) could reach 2000 km or more in regions with high topographic complexity, which means that the searching region is always in an irregular shape. Thus, for the area (e.g. the Tibetan Plateau and surrounding area) with complicated topography and multiplex hydroclimate variation, GLDD could identify the

unique searching region (including shape and size) rigorously for each target grid. While for the area with homogeneous hydroclimate variability and rainfall regime, GLDD could capture the proxies far from the target grid, which could reconstruct well at the areas where proxy data are not present, such as the east of the Caspian Sea.

**Figure 17:** The maximum (a) and minimum (b) distance from boundary of the searching region to the target point, and their difference (c) for each grid.

2) In Figs.  $5 \sim 8$ , the accuracy of the SPI reconstruction results in eastern China is very high. This is clearly because DWI is mainly used in eastern China. Using DWI is very important. However, unlike tree rings, the quality and quantity of the documentary material on which the DWI is based has been greatly improving as the times become newer, and from the 20th century, meteorological observation data itself should be included in the documentary materials. Since the calculations in Figs.  $5 \sim 8$  are based on the comparison of meteorological observation data and proxy data from 1948 CE, it is obvious that the SPI reconstruction accuracy will be higher in areas where DWI is used as a proxy than in areas where only tree rings are used, and it may not reflect the actual reconstruction accuracy of the SPI from 200 years ago or 300 years ago. There needs to be a mention of that possibility. In the same sense, since the observation data, it is obvious that the two coincide.

Accepted and revised. We discussed the issue in section 3.3 as following:

Thirdly, for documentary proxies, DW120 may use instrumental precipitation data to identify the dryness/wetness grades since 1951, especially after 1979, which might lead to overestimation of the calibration and verification metrics in eastern China. Fortunately, since the data of

dryness/wetness grades before 1950 are completely derived from historical documents (Wang and Zhao, 1979), comparison between the Fig. 12-13 and Figs 6-7 by each site-grid could help us to assess the overestimation, which shows that the overestimation of  $R^2a$  in this reconstructions is about 10% in average over eastern China (Line 368-374).

3) In this paper, calculations excluding tree-ring data that show negative correlations with precipitation are performed in Figs 5 and 7. In arid regions, there should be certainly a positive correlation between precipitation and tree ring width, but in humid regions, there is usually a negative relationship of precipitation with temperature and/or solar radiation, so it is rather common for there to be a negative correlation between precipitation and tree ring width. In fact, as shown in Figs 6 and 8, results using tree ring data that show negative correlations with precipitation are much better than excluding it. In other words, I do not understand the meaning of calculating Figs. 5 and 7.

Accepted and revised. This is why we reconstruct the SPI data with version A (for only including the positive correlation tree ring width as candidate proxies) and version B (for including all tree ring data as candidate proxies), and we also rewrite the related section to show this issue more clearly as followings:

According to the principle of dendroclimatology, soil water affects the growth rate and formation of wood, both within a season and the longer terms, thus, tree-ring width is expected to be positively correlated with precipitation via this direct response (Vaganov et al., 2011; Wettstein et al., 2011). For tree-ring density chronologies, they are usually correlated with temperature variation and scarcely used in precipitation reconstruction (Briffa et al., 2002). However, due to multiple types of climate and complex topography in the vast study area, the tree-ring density chronologies with negative correlations to precipitation may also well indicate precipitation variation (George, 2014) and the use of such tree-ring predictors in hydro-climate reconstruction has been discussed in previous studies (e.g. Cook et al., 2020). Therefore, we reconstruct two sets datasets of SPI, one excludes tree-ring width chronologies negatively correlated to precipitation and tree-ring density chronologies (hereafter called as "Version A"), the other includes all tree-ring chronologies (here after called "Version B"). (Line 126-135)

4) In this paper, the accuracy of the reconstruction of SPI in eastern China was improved by using DWI based on document materials. This is a very good thing, but there are a lot of documents related to weather since 1700 CE in Japan and elsewhere. In this paper, authors also use the oxygen isotope ratio of tree rings, which are highly correlated with precipitation. Since the data on the oxygen isotope ratio of tree rings has increased rapidly in recent years, it is expected that the results of this study will be further improved if calculations based on GLDD are performed by incorporating such document materials from other regions and data on the tree ring oxygen isotope ratio. It might be good to have such a comment in the text.

Accepted and revised. (1) We add the series of wet-season (May to October) rainy days for 5-site in Japan for reconstruction. The information is added in section 2.3 as follow:

In addition, the series of wet-season (May to October) rainy day for 5-site in Japan are also included. These series were extracted from the historical diaries (https://www.ncei.noaa.gov/access/paleo-search/study/5412) and merged with instrumental data (Kamiguchi et al., 2010) by the method from Murata (1992). (Line 158-161)

(2) For tree-ring oxygen isotope data, we have included all available chronologies from WDC-P and add some chronologies from recent publications. The related information is added in the section 2.3 as follows :

Tree-ring data are mainly (2772) from the International Tree Ring Data Bank (ITRDB), maintained by the World Data Center for Paleoclimatology (WDC-P, https://www.ncei.noaa.gov/products/paleoclimatology), including 1854 tree-ring width records, 828 tree-ring density records, 67 tree-ring latewood percent records, 22 tree-ring stable oxygen isotope ( $\delta^{18}$ O) and 1 tree-ring blue intensity record. (Line 104-106)

Besides ITRDB, 17 tree-ring width chronologies and 3 tree-ring  $\delta^{18}$ O chronologies that indicate local precipitation or drought from recently published papers are included in our study (Shah et al., 2007; Sass-Klaassen et al., 2008; Arsalani et al., 2018; Arsalani et al., 2015; Chen et al., 2016; Zhang et al., 2017; Pumijumnong et al., 2020; Xu et al., 2015; Buckley et al., 2017; Ukhvatkina et al., 2021; Akkemik et al., 2020; Kostyakova et al., 2017; Kucherov, 2010; Xu et al., 2013; Borgaonkar et al., 2010). (Line 119-125) Specific comments

Lines 95-101: In this paper, authors utilize raw tree-ring data, instead of using published tree ring chronologies, and process them according to author's own method. However, the reader cannot judge the effect of the data processing on the results. I would appreciate it if authors discuss in some way whether or not there is an impact from the processing of data?

Accepted and added. We utilize raw tree-ring data because published tree ring chronologies are standardized by various methods and their expressed population signal (EPS) values are not available. We use running mean technique (Altman et al., 2014) to identify disturbance and age-dependent splines (Melvin et al., 2007) to remove growth trend, which are commonly used methods in tree-ring studies and we do not modify them. We add discussions on the potential impact of tree-ring processing on the reconstruction in section 3.3 as following:

Second, for tree-ring proxies, we use the same standardization method to build chronologies when raw measurements are available. However, about 4.5% of the tree-ring proxies do not have raw measurement file thus we have to use the processed chronologies with various standardization methods by different data providers. As the test for some sites, the difference between chronologies could reach 20% in maximum from different standardization methods (Li et al., 2011). This may also induce the uncertainty in the reconstruction. (Line 364-368)

Line 174: What is the specific reason for the decision to limit the number of proxies used for calculations to 5?

Accepted and added. This is because that the sample length to develop calibration equations for reconstruction is about 50 years usually, and the sample size should be preferably 10 times (or more) the number of variables for BSR according to the principle of statistics (Sekaran, 2003). (line 209-211)

Lines 177-178: 1942 and 1943 are mistakes for 1742 and 1743. Accepted and revised. Figure 9: In this figure, the meaning of indicating the area where the correlation coefficient is negative can be understood. But the meaning of classifying the area where the correlation coefficient is positive by the p-value is difficult to understand. In fact, India's low p-values only mean that the period of meteorological observations there is long before 1948 CE, and I don't think it is relevant to the discussion in this figure.

Accepted and add. We rewrite that paragraph with adding information as follows:

(1) Noted that the length of instrumental data before 1948 varies for different weather station, i.e., the degree of freedom for calculating these correlations are different station by station, we show the significant level for all positive correlations station by station instead of the value of correlation coefficient directly with levels of  $p \le 0.01$ , 0.01 , <math>0.05 and <math>p > 0.1 (Fig. 12). (Line 284-287)

While the low p-values (i.e. p>0.1) of the positive correlation or negative correlation in part of stations (e.g., in India, southeastern Asia, etc.) might be induced by both uncertainties from reconstructions based on few proxies available and observations in early times, because the instrumental data from these station usually extend to the 1880 and before (e.g., several stations in India extend to 1836) with missing records and using defective rain gauge in early time frequently (GHCNm, 2022). (Line 294-297)